# Adapting Off-the-Shelf Speech Recognition Systems for Novel Words

Wiam Fadel [1,2,*], Toumi Bouchentouf [1], Pierre-André Buvet [2] and Omar Bourja [3]

1  Department of Applied Sciences Research Laboratory (LaRSA), Ecole Nationale des Sciences Appliquées d'Oujda, Mohammed First University, Oujda 60000, Morocco

2  Department of Théories Textes Numérique (TTN), Sorbonne Paris Nord University, 93430 Villetaneuse, France

3  Embedded Systems and Artificial Intelligence Department, Moroccan Foundation for Advanced Science, Innovation and Research, Rabat 10100, Morocco

*  Correspondence: wiam.fadel@ump.ac.ma

**Abstract:** Current speech recognition systems with fixed vocabularies have difficulties recognizing Out-of-Vocabulary words (OOVs) such as proper nouns and new words. This leads to misunderstandings or even failures in dialog systems. Ensuring effective speech recognition is crucial for the proper functioning of robot assistants. Non-native accents, new vocabulary, and aging voices can cause malfunctions in a speech recognition system. If this task is not executed correctly, the assistant robot will inevitably produce false or random responses. In this paper, we used a statistical approach based on distance algorithms to improve OOV correction. We developed a post-processing algorithm to be combined with a speech recognition model. In this sense, we compared two distance algorithms: Damerau–Levenshtein and Levenshtein distance. We validated the performance of the two distance algorithms in conjunction with five off-the-shelf speech recognition models. Damerau–Levenshtein, as compared to the Levenshtein distance algorithm, succeeded in minimizing the Word Error Rate (WER) when using the MoroccanFrench test set with five speech recognition systems, namely VOSK API, Google API, Wav2vec2.0, SpeechBrain, and Quartznet pre-trained models. Our post-processing method works regardless of the architecture of the speech recognizer, and its results on our MoroccanFrench test set outperformed the five chosen off-the-shelf speech recognizer systems.

**Keywords:** automatic speech recognition (ASR); out-of-vocabulary (OOV); French language; assistant robot





## 1. Introduction

The development of assistant robots that possess the capability of natural communication and the ability to exhibit empathy towards individuals, both at levels similar to that of a human, would represent a significant breakthrough in the field of robotics.

Robot assistants are primarily used as voice assistants with physical bodies [1]. A voice assistant is a system that uses a speech recognition module, a speech synthesis module, and a natural language processing (NLP) module to communicate like a human [2].

Ensuring effective speech recognition is crucial for the proper functioning of robot assistants. Non-native accents, new vocabulary, and aging voices can cause malfunctions in speech recognition systems. If this task is not executed correctly, then the voice assistant will inevitably produce false or random responses.

One essential subject of speech recognition systems is Out-of-Vocabulary words (OOV) [3]. OOV words are new terms that show up in the speech test set but do not exist in the recognition vocabulary. They are generally essential content terms, such as names and localities, that include information that is critical to the performance of several speech recognition tasks. Unfortunately, most speech recognition systems are closed-vocabulary and do not support OOV terms [4–6]. When a speech recognition system encounters a new term, it may mistake it for a similar sounding word in its vocabulary, leading to errors in

recognizing surrounding words. To address this, there is significant interest in developing speech recognition systems that can accurately detect and correct out-of-vocabulary words.

Current OOV research has focused on detecting the presence of OOV words in test data [7,8]. Solutions to this issue can be broadly categorized into two categories: detection-based approaches and correction-based approaches. The objective of detection-based approaches is to determine whether an error occurred in the transcription by utilizing features derived from the Automatic Speech Recognition (ASR) system, such as confidence scores, language model, and confusion network density [9]. Correction-based approaches, on the other hand, rely on the results of error detection to replace any misrecognized words with the correct ones.

Following the comparison of the most effective French speech recognition systems with the lowest error rates in a realistic setting [10], it was observed that many of these systems have difficulty recognizing speech within specific domains, such as in the medical field and concerning Moroccan culture. As a result, we aimed to investigate the current state of the art and devise ways to enhance the systems' capability to understand out-of-vocabulary words.

This paper is structured as follows. In Section 2, we review previous studies on enhancing speech recognition accuracy, including both statistical and deep learning approaches. Section 3 details the methodology employed in this study, including the corpus we constructed and its usage for testing purposes, a comparison of commonly used ASR systems, and a post-processing method for improving speech recognition for novel words. The results of our research are discussed in Section 4. Finally, in the conclusion, we summarize our findings and outline potential directions for future research.

## 2. Related Works on Improving Speech Recognition Accuracy

### 2.1. Statistical Approaches

These models use traditional statistical techniques such as n-grams, Hidden Markov Models (HMM), and specific linguistic rules to learn the probability distribution of words. They have been used in speech recognition systems for a long time, and they were found to be effective in many cases, especially when dealing with low-resource languages.

Kunisetti [11] tried various approaches to improve speech recognition accuracy for Telugu text. These approaches were (1) calculating distance using the Levenshtein distance algorithm and adding minimum distance variants to the static dictionary, (2) adding frequently occurring errors, (3) adding variants in the language model, (4) changing probabilities, and (5) modifying transcriptions. In all of these approaches, the author succeeded in achieving lower error rates when new variants were added to the dictionary.

In [12], the authors investigated the possibility of using sub-word vocabularies, where words are split into frequent and common parts. Sub-word vocabularies were extracted by performing word decomposition on the text corpus and taking the thousand most frequent units. For comparison, the full vocabulary of this study's corpus contained approximately 1.5 million surface forms. The authors explored two different methods: (1) fully unsupervised and data-driven word decomposition using the Morfessor tool and (2) word decomposition using a stemmer. The results of experiment (2) show that this allows for a significantly reduced OOV rate.

In [8], the authors presented the implementation of rule-based grapheme-to-phoneme G2P to assist the automatic generation of OOV word pronunciation for a speech recognition system. The proposed system gained an 87.29% overall phoneme error rate. Meanwhile, the achievable phoneme error rate for Indonesian words was 93.65%.

Pellegrini and his team examined the use of a Markov chain classifier with two states, error and correct, to model errors using a set of 15 common features in error detection. The system was tested on an American English broadcast news speech NIST corpus and correctly detected 860 errors with only a 16.7% classification error rate.

The paper [13] presented a two-stage technique for handling OOV words in speech recognition by using a mixed word/sub-word language model to transcribe OOV words

into sub-word units in the first stage and a look-up table to convert them to word hypotheses in the second stage. The technique was tested on out-of-domain data, and it was found to reduce the word error rate by 12% relative to a baseline system.

This article [14] addressed the problem of detecting errors in automatic transcriptions using statistical tools. The Markov chain model's ability to model temporal sequences was tested against a Gaussian mixture model and a maximum entropy model. Results show that the Markov chain model outperformed the other two, with a 16.7% CER and 860 errors correctly detected. The article suggested that the choice of using a Markov chain or maximum entropy model depends on the application.

The paper [15] presented a technique to improve ASR accuracy by using phonetic distance and domain knowledge to post-process the results of off-the-shelf speech recognition services. The authors used open-source Sphinx-based language models to decrease the WER of the Google speech recognition system. The results showed significant improvement over Google ASR and open-source ASR on various corpora, mainly from human–robot interaction.

The study conducted by Traum et al. [16] utilized a method of combining the results of both Google and Sphinx ASR services in order to achieve both general and domain-specific outcomes.

The authors [17] proposed a language model that is trained using a discriminative method and utilizes dependency parsing, which would allow for the utilization of long-distance structural features within sentences. The training process was done on a list of the top-scoring possibilities (n-best lists) using the perceptron algorithm. The model was then evaluated by reordering the top possibilities generated by recognizing speech from the Fisher data set, which contains informal telephone conversations. The results indicated that this approach of training with syntactic features using perceptron-based methods can lead to a decrease in the WER.

The paper [18] described a new approach to error correction in automatic speech recognition called SoftCorrect. SoftCorrect uses a soft error detection mechanism to avoid the limitations of both explicit and implicit error detection methods. It uses a probability produced with a dedicatedly designed language model to detect whether a token is correct or not. Experiments showed that SoftCorrect outperformed previous works by a large margin, achieving 26.1% and 9.4% CER reduction, respectively, while still enjoying fast parallel generation.

*2.2. Deep Learning Approaches*

On the other hand, deep learning approaches, such as neural networks, are based on the idea of learning representations of the input data. These models require large amounts of labeled data to train and learn the underlying structure of the speech input. However, once trained, they are able to extract high-level features from the speech signals, such as phonemes and sub-words, which makes them well suited to handling OOV words.

Sarma et al. [19] built an ASR error detector and corrector using co-occurrence analysis. They introduced a novel unsupervised approach for detecting and correcting misrecognized query words in a document collection. According to the authors, this method can produce high-precision targeted detection and correction of OOV words.

In the same context, Bassil and Semaan [20] proposed a post-editing ASR error correction method based on the Microsoft N-Gram data set for detecting and correcting spelling errors generated by ASR systems. The detection process consisted of detecting OOV word spelling errors in the Microsoft N-Gram data set, and the correction process consisted of two steps: the first one consisted of generating correction suggestions for the detected word errors, and the second one consisted of a context-sensitive error correction algorithm for selecting the best candidate for the correction. The error rate using the proposed method was around 2.4% on a data set composed of a collection of five different English articles, each with around 100 words read by different speakers.

More recently, [6] proposed a text processing model for Chinese speech recognition. It combined a bidirectional long short-term memory (Bi-LSTM) network and a conditional random field in two stages: text error detection and text error correction, respectively. Through verification and system tests using the SIGHAN 2013 Chinese Spelling Check (CSC) data set, the experimental outcomes indicated that the model can successfully enhance text following speech recognition accurately.

In [21], the authors proposed a machine translation-inspired sequence-to-sequence approach which learns to "translate" hypotheses to reference transcripts. To augment training data, the authors used all N-best hypotheses to form pairs with reference sentences, generated audio data using speech synthesis, and added noise to the source recordings. The resulting training set consisted of 640 M reference hypothesis pairs. The proposed system achieved an 18.6% relative WER (Word Error Rate) reduction.

Another paper [22] used a similar approach for Mandarin speech recognition but proposed a transformer model for spelling correction. The authors reported that it resulted in a 22.9% relative CER (Character Error Rate) improvement.

The paper [23] presented a sub-word modeling method for phoneme-level OOV word recognition in Amharic, using grapheme-to-phoneme conversion, syllabification for epenthesis, and sub-word-based decoding. The end-to-end models were trained and evaluated with a 22 h speech data set and a 5k testing data set. The experiment results showed that the phoneme-based BPE system with a syllabification algorithm was effective in achieving higher accuracy or minimum WER (18.42%) in the CTC-attention end-to-end method.

The paper [24] presented a method for handling OOV words in ASR systems that retrains an integral ASR system with a discriminative loss function and a decoding method based on a TG graph, using an open data set of the Russian language. The method aims to reduce the WER while maintaining the ability to recognize OOV words. Results showed that it reduced the WER by 3% compared to the standard method, making the system more resistant to recognizing new unseen words.

The paper [25] presented FastContext, a method for handling out-of-vocabulary (OOV) words in natural language processing systems. It improved the embedding of sub-word information by using context-based embedding computed with a deep learning model. The method was evaluated on tasks of word similarity, named entity recognition, and part-of-speech tagging, and it performed better than FastText and other state-of-the-art OOV-handling techniques. The results indicate that the approach was able to capture semantic or morphological information, and it was more effective when the context was the most relevant source to infer the meaning of the OOV words.

The paper [26] described the THUEE team's approach for handling Out-of-Vocabulary (OOV) words in their speech recognition system for the IARPA Open Automatic Speech Recognition Challenge (OpenASR21). They used Grapheme-to-Phoneme (G2P) techniques to extend the pronunciation lexicon for OOV and potential new words in the Constrained training condition. They also applied multiple data augmentations techniques. In the Constrained-plus training condition, they used the self-supervised learning framework wav2vec2.0 and experimented with various fine-tuning techniques with the Connectionist Temporal Classification (CTC) criterion on top of the publicly available pre-trained model XLSR-53. They found that using the CTC model fine-tuned in the target language as the frontend feature extractor was effective in improving its OOV-handling performance.

Because deep learning methods require more data, we will start by explaining an effective statistical approach. Statistical techniques can be better than modern techniques for handling OOV words in speech recognition for low-resource languages because they require less data to train, are more interpretable, and can be a good fit for languages with simple grammar and orthographic structures.

## 3. Methodology

### 3.1. Data Set

We gathered audio recordings and their corresponding texts to use as a test set for our proposed method. The test set was composed of the same content as the MoroccanFrench corpus [10], but with new voices. It included recordings that depicted Moroccan culture as well as scenarios related to the COVID-19 pandemic. The test set included 100 audio recordings and their accompanying texts, which were recorded by 10 speakers of Moroccan French. Each recording was between 5 and 7 s in length. These recordings were made in an office setting with background noise and using computer microphones. Table 1 provides a sample from the test set.

**Table 1.** Extract from the Test set.

| Audio_Path | Text |
|---|---|
| /audios/1.wav | quelles sont les communautés touchées par la pandémie de COVID-19 ? |
| /audios/2.wav | chefchaoun n'est pas très loin de tanger |
| /audios/3.wav | les marocains seront vaccinés avec sinopharm |

### 3.2. Speech Recognition System

In this study, we evaluated the performance of various off-the-shelf speech recognition systems, namely Google speech-to-text API, VOSK API, QuartzNet, Wav2vec2.0, and CRDNN pre-trained model, on the MoroccanFrench corpus. The results, as shown in Table 2, indicate that the Google speech-to-text API had the lowest average error rate, with a WER of 0.38. The other ASR systems, however, had relatively high error rates. This is likely due to the fact that these systems were trained on general French language, whereas the audio content in the MoroccanFrench corpus was sourced from the Moroccan culture and COVID-19 domains.

**Table 2.** WER results of ASR systems on the MoroccanFrench corpus.

| ASR | Google API | Vosk API | QuartzNet | Wav2vec2.0 | CRDNN |
|---|---|---|---|---|---|
| WER | 0.38 | 0.57 | 0.48 | 0.55 | 0.47 |

It is important to note that background noise and non-native accents in the tested audio recordings can have a significant impact on the performance of ASR systems. As the audio in the MoroccanFrench corpus contains both of these factors, it was a challenging test set for the ASR systems. The Google speech-to-text API was able to handle these challenges better than the other ASR systems, resulting in the lowest WER. It was able to recognize the word "COVID-19" because its model is still being updated. Moreover, the corpus includes some commonly known abbreviations, such as "WHO" (World Health Organization), which were only recognized by using a speech-to-text system. These findings lead us to think about adapting speech recognition models to our domain.

It is worth mentioning that we are working on a robot assistant that will be deployed in an open space in Morocco. This environment will be exposed to different non-native accent and background noise, which will affect the performance of the ASR system. Therefore, it is crucial to have a robust system that can handle these challenges.

### 3.3. Speech Recognition System Post-Processing

Our goal was to improve the accuracy of speech recognition systems by correcting errors in the text output without making changes to the underlying system architecture. To achieve this, we used a statistical method based on Levenshtein distance, which can work with less data and is specific to a certain field. This method is particularly useful in situations where adjustments to the speech recognition system are not possible, such as when the system is a "black box" product.

We measured word similarity using Levenshtein distance and its extension (Damerau–Levenshtein-DL). It is often used for spelling correction. It calculates word similarity according to the minimal number of basic operations required to get to the target word from the original word.

### 3.3.1. Levenshtein Algorithm

The algorithm introduced by Levenshtein measures the similarity between two words by calculating an edit distance. The edit distance is defined as the minimum number of basic editing operations needed to transform a wrong word into a dictionary word. Thus, to correct a wrong word, the method retains a set of solutions requiring the fewest possible editing operations [27].

Mathematically, the Levenshtein distance between two strings a and b is given as lev(a,b), where the indicator function is equal to 0 when ai = bj and equal to 1 otherwise, and where leva, b(i,j) is the distance between the first i characters of a and the first j characters of b.

$$lev_{a,b}(i,j) = \begin{cases} \max(i,j) & \text{If} \min(i,j) = 0, \\ \min \begin{cases} lev_{a,b}(i-1,j) + 1 \\ lev_{a,b}(i,j-1) + 1 \\ lev_{a,b}(i-1,j-1) + 1_{(ai \neq bj)} \end{cases} & \text{otherwise.} \end{cases} \tag{1}$$

Equation (1) Levenshtein algorithm.

### 3.3.2. Damerau–Levenshtein Algorithm

The Damerau–Levenshtein distance differs from the classical Levenshtein distance by including transpositions among its allowable operations in addition to the three classical single-character edit operations (insertions, deletions, and substitutions). Damerau defined a kind of model for spelling errors. His method fixed four types of errors: insertion, deletion, substitution, and permutation.

Correcting spelling errors involves comparing a misspelled word to the words in a dictionary for a specific language and suggesting a list of words that are most similar to the misspelled word. This is done by measuring the similarity and distance between the words. In order to use Levenshtein distance, we created a custom vocabulary by linking each correctly spelled out-of-vocabulary (OOV) word with its corresponding misspelled versions, as shown in Table 3.

**Table 3.** Types of misspelled words.

| OOV Word | Misrecognized Words | Type of Mistake |
|----------|---------------------|-----------------|
| COVID-19 | Couvi | Word reduction |
|          | covite dix-neuv | Missing/misspelled symbols |
|          | lacovitedineuf | Multiple words separation |
| mascir   | Massir | misspelled symbols |
|          | Massyr | misspelled symbols |
|          | Macir | misspelled symbols |

After speech recognition, we tokenized the input text into individual words (tokens). We then compared each word to the out-of-vocabulary (OOV) words that were misrecognized. To determine how similar the words were, we used Levenshtein distance. If the similarity score between two words was low, it means that the word in the text was detected as an OOV, and we would then replace it with the correct OOV word.

The proposed method is a post-correction technique for the outputs of speech recognition systems. It utilizes a combination of dictionary lookup and the Levenshtein distance measure to correct any potential misspellings. The method consists of three steps:

- Dictionary Creation: The first step is to create a dictionary that includes both the misspelled words and their corresponding correct words, with the correct word being stored as a keyword in the dictionary.
- Searching for a Match: In this step, the word of interest (the output from the ASR) is searched within the dictionary values. If the word is found, then it is considered a misspelling. If not, then it is considered to be spelled correctly.
- Calculation of Levenshtein Distance: If a misspelling is identified in the previous step, then the word of interest is then subjected to the Levenshtein distance measure. This measure calculates the distance between the misspelled word and all words in the dictionary. The correct word is chosen as the keyword with the smallest Levenshtein distance, as this indicates the highest similarity between the two words.

The goal of correction is to replace any inaccurate word with its closest match from the dictionary. Our method utilizes traditional techniques for determining the similarity between strings of text, such as Levenshtein distance. To evaluate the performance of this approach, we recalculated the Word Error Rate (WER) of ASR systems using this method, as illustrated in Figure 1.

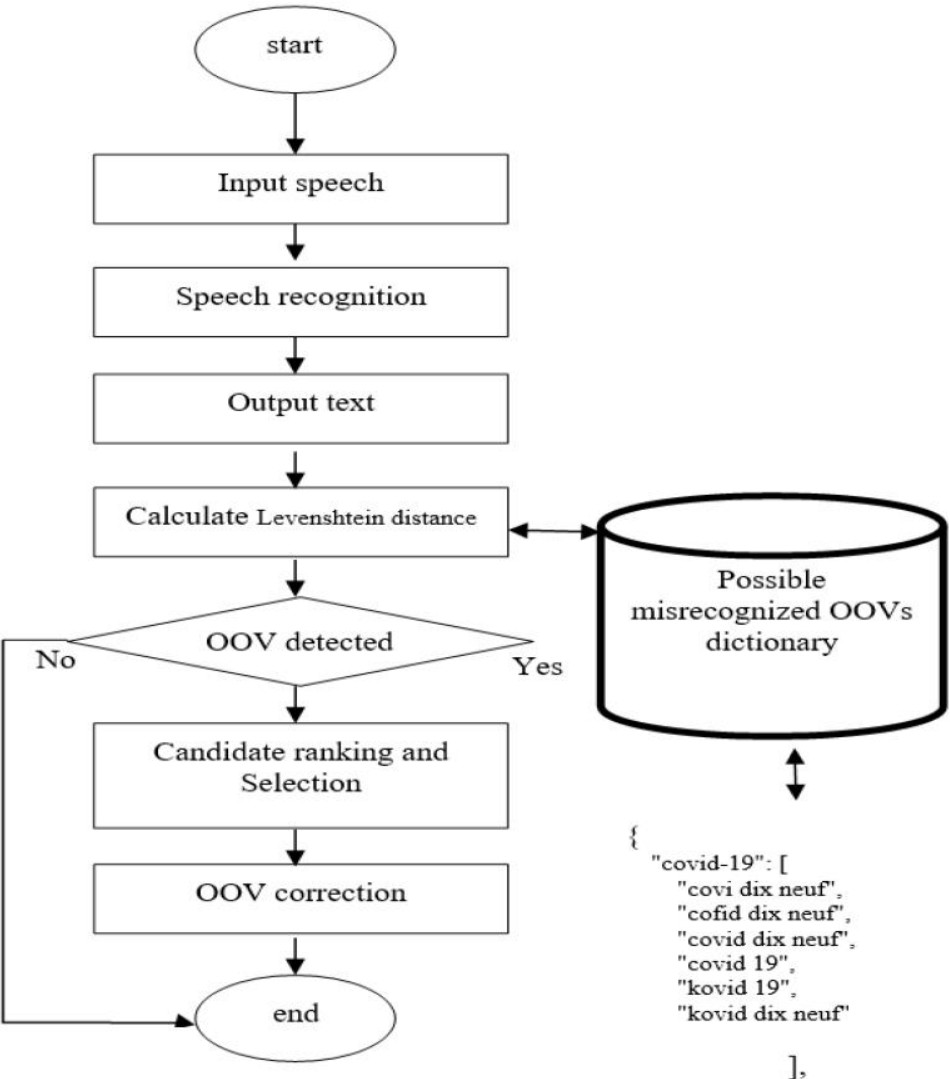

**Figure 1.** Speech recognition post-processing.

## 4. Results and Discussion

Table 4 shows the results of evaluating different automatic speech recognition (ASR) systems on the MoroccanFrench test set, both before and after post-processing. The table compares the word error rate (WER) of each ASR system before and after post-processing, and it also shows a comparison between two different algorithms of post-processing, Levenshtein and Damerau–Levenshtein. WER is a measure of the accuracy of the transcription, with a lower WER indicating better performance.

**Table 4.** Comparison of this method to different off-the-shelf ASR systems.

| ASR Systems | WER % before Post-Processing | WER % after Processing | | | |
|---|---|---|---|---|---|
| | | Levenshtein | | Damerau Levenshtein | |
| | | min = 0.2 | max = 0.8 | max = 0.85 | max = 0.9 |
| Speech-to-text | 38.27 | **37** | 38.37 | 38.36 | 37.72 |
| Wav2vec 2.0 | 55.96 | 51.71 | 51.26 | 53.08 | 51.71 |
| Quartznet | 48.35 | 48.04 | 47.08 | 47.44 | 47.81 |
| SpeechBrain | 47.03 | 45.98 | 45.63 | 45.81 | 45.93 |
| VOSK | 57.96 | 57.52 | 57.82 | 57.85 | 57.52 |

The results in the table indicate that the WER of the ASR systems was high before post-processing. For example, the WER of the speech-to-text system was 38.27%, that of the Wav2vec 2.0 system was 55.96%, and that of the Quartznet system was 48.35%. This was due to the specific data set used for evaluation, which is designed to simulate real-world scenarios with noise, different accents, and new vocabulary, making transcription more difficult.

After post-processing, the WER of the ASR systems was reduced, indicating that post-processing improved the accuracy of the transcription. For example, the WER of the speech-to-text system was reduced to 37%, that of the Wav2vec 2.0 system was reduced to 51.71%, and that of the Quartznet system was reduced to 48.04%.

The post-processing method used in this evaluation is based on the Levenshtein and Damerau–Levenshtein algorithm. The threshold for the Levenshtein algorithm was set to the minimum, where a threshold of 0.2 means that two words are similar. On the other hand, for the Damerau algorithm, the threshold was set to the maximum, where a threshold of 0.8, 0.85, or 0.9 means that two words are similar.

Wav2vec 2.0 was able to significantly reduce the error rate due to its use of an acoustic model without a language model, which resulted in the generation of words that were phonetically correct but had incorrect orthography. This made our approach more effective. Our tests indicate that speech recognition systems that are paired with a language model trained in various domains may produce lexically correct but contextually incorrect predictions (as seen in Table A1 of Appendix A). For instance, a simple sentence predicted by a speech-to-text system that has not been trained on the word "mascir" may make predictions based on acoustic similarities and identify it as a phonetically matching in-vocabulary item, such as in the case of "mascir" and "ma soeur" depicted below.

♦     Mascir relève de l'unsiversité Mohammed VI polytechnique.
♦     Ma sœur relève de l'université Mohammed VI polytechnique.

This prompted us to consider incorporating a language model that comprehends the context of the sentence for more accurate results.

Our results align with other studies [6,13], that have demonstrated the effectiveness of post-processing methods in reducing errors for recognizing out-of-vocabulary (OOV) words in automatic speech recognition (ASR) systems. However, in contrast, our experiment using the method of Ali et al. [28] with different scoring functions showed poor results. Utilizing a dictionary with only one misspelled word and its correct form, we discovered

that their method is unable to handle various types of errors within a single word, such as those caused by different accents. In comparison, our method improves upon this by pairing each correctly spelled OOV word with its potential misspelled versions, allowing for more accurate matching of misspelled OOV words to their correct forms, a task that proves difficult for the method of Ali et al. Additionally, machine learning techniques generally need a large amount of training data to correct the misspelled OOV word with high accuracy [17,19,20], and constructing training data by hand is costly. As compared with them, our method can be realized with low cost. Moreover, our method does not depend on particular speech recognizers [6,19,20].

## 5. Conclusions

In this work, we demonstrate the effectiveness of using a statistical approach to reduce errors in speech recognition systems. The Levenshtein and Damerau algorithms were compared by varying a threshold to minimize the WER on the MoroccanFrench test set. The results showed that the Damerau algorithm was more effective than the Levenshtein algorithm in reducing the WER, with a threshold of max = 0.8.

Our results are in line with other studies [6,13] that have reported the effectiveness of post-processing methods in reducing errors for recognizing OOV words in ASR systems.

The post-processing method was used to detect and correct OOV words in the text after the speech recognition task. In terms of time consumption, post-processing took only 0.029 s. In terms of WER reduction, it was between 1% and 4%, which indicates the method succeeded in reducing the error without modifying the ASR system itself. However, there is still room for improvement in this method.

In addition to the points mentioned previously, it is worth noting that the findings of this research have important theoretical and practical implications for the field of speech recognition. Theoretically, this research demonstrates the effectiveness of using statistical approaches to reduce errors in off-the-shelf speech recognition systems, which can inform future research in this area. Practically, the results of this research have potential applications in real-world scenarios, such as speech-controlled medical assistant robots that can be implemented in retirement homes.

However, it is important to note that there are limitations to this research. One limitation is that the data set used for evaluation had a relatively low vocabulary of possible misrecognized OOV words. Additionally, the testing data set was noisy and relatively small, which may limit the generalizability of the results.

To address these limitations, we will expand the data set to include more diverse accents and languages, and we will explore the use of other post-processing algorithms and machine learning techniques to improve the accuracy of speech recognition systems.

**Author Contributions:** Conceptualization, W.F.; methodology, W.F.; software, W.F.; validation, T.B. and P.-A.B.; formal analysis, W.F.; investigation, W.F., T.B., P.-A.B. and O.B.; resources, O.B.; data curation, W.F.; writing—original draft preparation, W.F.; writing—review and editing, W.F.; supervision, T.B., P.-A.B. and O.B.; project administration, O.B. All authors have read and agreed to the published version of the manuscript.

**Funding:** This research received no external funding.

**Institutional Review Board Statement:** Not applicable.

**Informed Consent Statement:** Not applicable.

**Data Availability Statement:** Currently, the "MoroccanFrench" corpus data set is not publicly available. It consists of 500 audio recordings in the form of WAV files as well as their corresponding transcriptions. Access to this data set may be granted upon request and with the approval of the research project leader.

**Acknowledgments:** We would like to acknowledge the support and contributions of Teamnet and Kompai Robotics as our industrial partners in the "Medical Assistant Robot" project. This project is a collaboration between MAScIR (Moroccan Foundation for Advanced Science, Innovation and Research), ENSIAS (Ecole Nationale Supérieure d'Informatique et d'Analyse des Systèmes), ENSAO (Ecole Nationale des Sciences Appliquées d'Oujda), and USPN (Université de Sorbonne Paris Nord), and it has been supported by the Moroccan Ministry of Higher Education, Scientific Research and Innovation, the Digital Development Agency (DDA), and the National Center for Scientific and Technical Research of Morocco (CNRST) through the "Al-Khawarizmi project". The project code is Alkhawarizmi/2020/15.

**Conflicts of Interest:** The authors declare no conflict of interest.

## Appendix A

**Table A1.** Speech recognition examples from various ASR systems on Moroccan French accents.

| ASR System | Recognized Text with Different Accents:<br>« COVID-19 a Impact Important Sur la Production D'aliments » |
| --- | --- |
| Speech-to-text API | mercredi 19 a un impact sur la protection d'allah |
| | la copine de 19 a un impact important sur la production d'aliments |
| | ma COVID-19 a un impact important sur la production d'aliments |
| VOSK API | la mauvaise dix-neuf a un impact important sur jim protection d'allure |
| | la convives dix-neuf a un impact important sur la production d'aliments |
| | la coville dix-neuf un impact important sur la production d'aliments |
| Wav2vec2.0 | lacovdnua un inpacinpton sur la protection d'al |
| | la covitisnetea un impacte important sur la production d'alément |
| | la copitisneur e l'ipote dmportant sur leu production d'alim |
| QuartzNet | l'alcrovie dix neuf a un impact important sur la production d'alian |
| | la courbide est emportant sur la production d'als |
| | la couvite a un impact important sur la production d'animaux |
| SpeechBrain | la crevée dix neuf a à un bac important sur la protection d'allemagne |
| | la covie dix neuf a un impact important sur la production d'aligny |
| | la courbie dix neuf a un impact important sur la production d'alléments |

This table compares the text recognized by different speech recognition systems when used with different accents. The original text is "COVID-19 a un impact important sur la production d'aliments". The different columns represent different speech recognition systems, and the different rows represent the different versions of the text recognized by each system. Overall, the different systems had difficulty correctly recognizing the text when used with different accents.

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
