# Peer review of "Adapting Off-the-Shelf Speech Recognition Systems for Novel Words"

_information, doi:10.3390/info14030179_

Round 1

Author Response

Response to Reviewer 1 Comments

Point 1:
Introduction
It would be helpful to provide reasoning for including the “other works” that you do and how they relate to your study. 

Response 1: 

Section: 2. Related works on improving speech recognition accuracy

Statistical approaches   

These models use traditional statistical techniques such as n-grams, Hidden Markov Models (HMM), and specific linguistic rules to learn the probability distribution of words. They have been used in speech recognition systems for a long time, and they have been found to be effective in many cases, especially when dealing with low-resource languages. 

.... 

 Deep Learning approaches (instead of "other works"):

On the other hand, deep learning approaches, such as neural networks, are based on the idea of learning representations of the input data. These models require large amounts of labeled data to train and learn the underlying structure of the speech input. But once trained, they are able to extract high-level features from the speech signals, such as phonemes and subwords, which makes them well suited to handling OOV words.

....

Since deep learning methods require more data, we will start with an effective statistical approach. statistical techniques can be better than modern techniques for handling OOV words in speech recognition for low-resource languages because they require less data to train, are more interpretable, and can be a good fit for languages with simple grammars and orthographic structures

Point2: At the end of the introduction, it would be helpful to remind the reader of the purpose of your study.

Response 2: 

Our goal is to develop a voice assistant for the elderly residents of a French retirement home in Rabat, Morocco by implementing the appropriate algorithms. The success of the assistant relies on having a speech recognition system that is able to handle variations in accent, such as the French-Moroccan accent, as well as new vocabulary (OOV words) and the voices of older individuals. These variations can cause issues with the performance of the assistant, and if the speech recognition is not done correctly, the voice assistant may provide incorrect or unexpected responses.

For this reason, we have rephrased our introduction with the following:

Section 1: Introduction:

The development of assistant robots that possess the capability of natural communication and the ability to exhibit empathy towards individuals, similar to that of a human, would represent a significant breakthrough in the field of robotics.

The robot assistant is primarily used as a voice assistant with a physical body [1]. A voice assistant is a system that uses a speech recognition module, a speech synthesis module, and a natural language processing (NLP) module to communicate like a human [2].

Ensuring effective speech recognition is crucial for the proper functioning of robot assistants. Non-native accents, new vocabulary, and aging voices can cause malfunctioning of the speech recognition system. If this task is not executed correctly, the voice assistant will inevitably produce false or random responses.

One essential subject of speech recognition systems is Out-of-Vocabulary words (OOV) [3]. OOV words are new terms that show up in the speech test set but do not exist in the recognition vocabulary. They are generally essential content terms like names and localities that include information that is critical to the performance of sev-eral speech recognition tasks. Unfortunately, most speech recognition systems are closed vocabulary and do not support OOV terms [4][5], [6]. When a speech recogni-tion system encounters a new term, it may mistake it for a similar sounding word in its vocabulary, leading to errors in recognizing surrounding words. To address this, there is a significant interest in developing speech recognition systems that can accurately detect and correct out-of-vocabulary words.

Current OOV research has focused on detecting the presence of OOV words in test data [7] [8]. Solutions to this issue can be broadly categorized into two categories: detection-based approaches and correction-based approaches. The objective of detec-tion-based approaches is to determine whether an error has occurred in the transcrip-tion by utilizing features derived from the automatic speech recognition (ASR) system, such as confidence scores, language model, and confusion network density [9]. Correc-tion-based approaches, on the other hand, rely on the results of error detection to re-place any misrecognized words with the correct ones.

Following the comparison of the most effective French speech recognition systems with the lowest error rate in a realistic setting [10], it has been observed that many of these systems have difficulty recognizing speech within specific domains such as the medical field and Moroccan culture. As a result, we aim to investigate the current state of the art and devise ways to enhance the systems' capability to understand out-of-vocabulary words.

This paper is structured as follows. In Section 2, we review previous studies on enhancing speech recognition accuracy, including both statistical and deep learning approaches. Section 3 details the methodology employed in this study, including the corpus we constructed and its usage for testing purposes, a comparison of commonly used ASR systems, and post-processing method for improving speech recognition for novel words. The results of our research are discussed in Section 4. Finally, in the conclusion, we summarize our findings and outline potential directions for future research.

Point 3:

Results
Can you compare your results better to previous literature? 

Response 3 :

As we were the first ones to try combining distance algorithms with commonly used speech recognition systems such as Google Speech-to-text, Vosk, SpeechBrain, QuartzNet, and Wav2Vec, we were unable to compare our results to other studies as we used our own test set for evaluation. Nevertheless, our findings are consistent with those of other research that have similarities to our approach.

Section: 5. Conclusion

 In this work, we demonstrate the effectiveness of using a statistical approach to reduce errors in speech recognition systems. The Levenshtein and Damerau algorithms were compared by varying a threshold to minimize the WER on MoroccanFrench test set. The results showed that the Damerau algorithm was more effective than the Levenshtein algorithm in reducing the WER, with a threshold of max=0.8.

Our results are in line with other studies [14], [22] that have reported the effective-ness of post-processing methods in reducing errors for recognizing OOV words in ASR systems.

The post-processing method was used to detect and correct OOV words in the text after the speech recognition task. In terms of time consumption, the post-processing took only 0.029 seconds. In terms of WER reduction, it was between 1% and 4%, which indicates the method succeeded in reducing the error without modifying the ASR system itself. However, there is still room for improvement in this method.

In addition to the points mentioned previously, it is worth noting that the findings of this research have important theoretical and practical implications for the field of speech recognition. Theoretically, this research demonstrates the effectiveness of using statistical approaches to reduce errors in off-the-shelf speech recognition systems, which can inform future research in this area. Practically, the results of this research have potential applications in real-world scenarios such as speech-controlled medical assistant robots that can be implemented in retirement homes.

However, it is important to note that there are limitations to this research. One limitation is that the dataset used for evaluation has a relatively low vocabulary of pos-sible misrecognized OOV words. Additionally, the testing dataset is noisy and relative-ly small, which may limit the generalizability of the results.

To address these limitations, we will expand the dataset to include more diverse accents and languages, and to explore the use of other post-processing algorithms and machine learning techniques to improve the accuracy of speech recognition systems.

 References

  1. Pollmann, C. Ruff, K. Vetter, and G. Zimmermann, “Robot vs. voice assistant: Is playing with pepper more fun than playing with alexa?,” ACM/IEEE International Conference on Human-Robot Interaction, pp. 395–397, Mar. 2020, doi: 10.1145/3371382.3378251. 
  2. ​“How to Build Domain Specific Automatic Speech Recognition Models on GPUs | NVIDIA Developer Blog.” https://developer.nvidia.com/blog/how-to-build-domain-specific-automatic-speech-recognition-models-on-gpus/ (accessed Nov. 29, 2021). 
  3. ​T. Desot, F. Portet, and M. Vacher, “End-to-End Spoken Language Understanding: Performance analyses of a voice command task in a low resource setting,” Comput Speech Lang, vol. 75, p. 101369, Sep. 2022, doi: 10.1016/J.CSL.2022.101369. 
  4. ​J. Kim and P. Kang, “K-Wav2vec 2.0: Automatic Speech Recognition based on Joint Decoding of Graphemes and Syllables,” arXiv:2110.05172v1, 2021. 
  5. ​A. Laptev, A. Andrusenko, I. Podluzhny, A. Mitrofanov, I. Medennikov, and Y. Matveev, “Dynamic Acoustic Unit Augmentation with BPE-Dropout for Low-Resource End-to-End Speech Recognition,” Sensors 2021, Vol. 21, Page 3063, vol. 21, no. 9, p. 3063, Apr. 2021, doi: 10.3390/S21093063. 
  6. ​A. Y. Andrusenko and A. N. Romanenko, “Improving out of vocabulary words recognition accuracy for an end-to-end Russian speech recognition system,” Scientific and Technical Journal of Information Technologies, Mechanics and Optics, vol. 22, no. 6, pp. 1143–1149, Nov. 2022, doi: 10.17586/2226-1494-2022-22-6-1143-1149. 
  7. ​J. v. Lochter, R. M. Silva, and T. A. Almeida, “Multi-level out-of-vocabulary words handling approach,” Knowl Based Syst, vol. 251, Sep. 2022, doi: 10.1016/J.KNOSYS.2022.108911. 
  8. ​F. Y. Putri, D. Hoesen, and D. P. Lestari, “Rule-Based Pronunciation Models to Handle OOV Words for Indonesian Automatic Speech Recognition System,” Proceeding - 2019 5th International Conference on Science in Information Technology: Embracing Industry 4.0: Towards Innovation in Cyber Physical System, ICSITech 2019, pp. 246–251, Oct. 2019, doi: 10.1109/ICSITECH46713.2019.8987472.
  9. Errattahi, A. El Hannani, and H. Ouahmane, “Automatic speech recognition errors detection and correction: A review,” Procedia Computer Science, vol. 128, pp. 32–37, 2018, doi: 10.1016/J.PROCS.2018.03.005.
  10. ​ Fadel, I. Araf, T. Bouchentouf, P.-A. Buvet, F. Bourzeix, and O. Bourja, “Which French speech recognition system for assistant robots?,” pp. 1–5, Mar. 2022, doi: 10.1109/IRASET52964.2022.9737976. 
  11. Subramanyam, “Improving Speech Recognition Accuracy Using Levenshtein Distance algorithm”.
  12. Salimbajevs, “Using sub-word n-gram models for dealing with OOV in large vocabulary speech recognition for Latvian”.
  13. Y. Putri, D. Hoesen, and D. P. Lestari, “Rule-Based Pronunciation Models to Handle OOV Words for Indonesian Automatic Speech Recognition System,” Proceeding - 2019 5th International Conference on Science in Information Technology: Embracing Industry 4.0: Towards Innovation in Cyber Physical System, ICSITech 2019, pp. 246–251, Oct. 2019, doi: 10.1109/ICSITECH46713.2019.8987472.
  14. Réveil, K. Demuynck, and J. P. Martens, “An improved two-stage mixed language model approach for handling out-of-vocabulary words in large vocabulary continuous speech recognition,” Comput Speech Lang, vol. 28, no. 1, pp. 141–162, Jan. 2014, doi: 10.1016/J.CSL.2013.04.003. 
  15. ​[12] Pellegrini and I. Trancoso, “Error detection in broadcast news ASR using Markov chains,” Lecture Notes in Computer Science (including subseries Lecture Notes in Artificial Intelligence and Lecture Notes in Bioinformatics), vol. 6562 LNAI, pp. 59–69, 2011, doi: 10.1007/978-3-642-20095-3_6/COVER. 
  16. Sarma and D. D. Palmer, “Context-based Speech Recognition Error Detection and Correction”.
  17. Bassil and P. Semaan, “ASR Context-Sensitive Error Correction Based on Microsoft N-Gram Dataset,” Mar. 2012, doi: 10.48550/arxiv.1203.5262.
  18. Yang, Y. Li, J. Wang, and Z. Tang, “Post Text Processing of Chinese Speech Recognition Based on Bidirectional LSTM Networks and CRF,” Electronics 2019, Vol. 8, Page 1248, vol. 8, no. 11, p. 1248, Oct. 2019, doi: 10.3390/ELECTRONICS8111248.
  19. “(2) (PDF) Post-editing and Rescoring of ASR Results with Edit Operations Tagging.” https://www.researchgate.net/publication/351687519_Post-editing_and_Rescoring_of_ASR_Results_with_Edit_Operations_Tagging (accessed Jul. 25, 2022).
  20. Zhang, M. Lei, and Z. Yan, “Automatic Spelling Correction with Transformer for CTC-based End-to-End Speech Recognition,” Mar. 2019, doi: 10.48550/arxiv.1904.10045.
  21. D. Emiru, S. Xiong, Y. Li, A. Fesseha, and M. Diallo, “Improving amharic speech recognition system using connectionist temporal classification with attention model and phoneme-based byte-pair-encodings,” Information (Switzerland), vol. 12, no. 2, pp. 1–22, Feb. 2021, doi: 10.3390/INFO12020062.
  22. Y. Andrusenko and A. N. Romanenko, “Improving out of vocabulary words recognition accuracy for an end-to-end Russian speech recognition system,” Scientific and Technical Journal of Information Technologies, Mechanics and Optics, vol. 22, no. 6, pp. 1143–1149, Nov. 2022, doi: 10.17586/2226-1494-2022-22-6-1143-1149. 
  23. ​R. M. Silva, J. v. Lochter, T. A. Almeida, and A. Yamakami, “FastContext: Handling Out-of-Vocabulary Words Using the Word Structure and Context,” Lecture Notes in Computer Science (including subseries Lecture Notes in Artificial Intelligence and Lecture Notes in Bioinformatics), vol. 13654 LNAI, pp. 539–557, 2022, doi: 10.1007/978-3-031-21689-3_38. 
  24. ​J. Zhao et al., “The THUEE System Description for the IARPA OpenASR21 Challenge,” Proceedings of the Annual Conference of the International Speech Communication Association, INTERSPEECH, vol. 2022-September, pp. 4855–4859, 2022, doi: 10.21437/INTERSPEECH.2022-269. 
  25. L. Aouragh, A. Yousfi, U. Sidi, M. Ben Abdellah, and G. Hicham, “Adapting the Levenshtein Distance to Contextual Spelling Correction,” International Journal of Computer Science and Applications, ÓTechnomathematics Research Foundation, vol. 12, no. 1, pp. 127–133, 2015.

Reviewer 2 Report

Poor English. Many grammatical and spelling errors.

Table 4 shows that found word error reduction is on the scale of experimental noise.

The conclusion should as follows: "described method looks inadequate to the OOV task"

Author Response

Response to Reviewer 2 Comments

Point 1: 

Poor English. Many grammatical and spelling errors.

Response 1: 

we have attempted to clarify the text in all sections by making corrections.

Point 2: 

Table 4 shows that found word error reduction is on the scale of experimental noise.

Response 2: 

We mention in the discussion that Table 4 shows that the found word error rate is higher on the scale of experimental noise, indicating that the results align with real-world scenarios, where the model or system's performance may not be optimal.

Section: 4. Results and Discussion (new version)

The table 4 shows the results of evaluating different automatic speech recognition (ASR) systems on MoroccanFrench test set, both before and after post-processing. The table compares the word error rate (WER) of each ASR system before and after post-processing, and also show a comparison between two different algorithms of post-processing, Levenshtein and Damerau Levenshtein. WER is a measure of the accuracy of the transcription, with a lower WER indicating better performance.

The results in the table indicate that the WER of the ASR systems is high before post-processing. For example, the WER of the Speech-to-text system is 38.27%, the Wav2vec 2.0 system is 55.96%, and the Quartznet system is 48.35%. This is due to the specific dataset used for evaluation, which is designed to simulate real-world scenarios with noise and different accents and new vocabulary, making the transcription more difficult.

After post-processing, the WER of the ASR systems is reduced, indicating that post-processing improves the accuracy of the transcription. For example, the WER of the Speech-to-text system is reduced to 37%, the Wav2vec 2.0 system is reduced to 51.71%, and the Quartznet system is reduced to 48.04%.

The post-processing method used in this evaluation is based on the Levenshtein and Damerau Levenshtein algorithm. The threshold for the Levenshtein algorithm is set to the minimum, where a threshold of 0.2 means that two words are similar. On the other hand, for the Damerau algorithm, the threshold is set to the maximum, where a threshold of 0.8, 0.85 or 0.9 means that two words are similar.

Wav2vec 2.0 was able to significantly reduce the error rate due to its use of an acoustic model without a language model, which resulted in the generation of words that were phonetically correct but had incorrect orthography. This made our approach more effective. Our tests indicate that speech recognition systems that are paired with a language model trained in various domains may produce lexically correct but contextually incorrect predictions (as seen in Table 5). For instance, a simple sentence predicted by a speech-to-text system that has not been trained on the word "mascir" may make predictions based on acoustic similarities and identify it as a phonetically matching in-vocabulary item, such as in the case of "mascir" and "ma soeur".

  • Mascir relève de l’unsiversité Mohammed VI polytechnique.
  • Ma sœur relève de l’université Mohammed VI polytechnique.

This led us to think of adding a language model that understands the context of the sentence for a more correct result.

Point 3: 

The conclusion should as follow: "described method looks inadequate to the OOV task"

Response 3: 

We mention in the discussion that our approach focuses on recognizing out-of-vocabulary (OOV) words after speech recognition. However, due to the nature of the dataset used which contains noise, new vocabulary, and non-native French accents, as well as the small vocabulary of misrecognized OOVs, we have observed a low significant improvement in the test dataset. To accurately evaluate the performance of our approach in real-world scenarios, we plan to test it on similar data, such as in a retirement home in Morocco, where our medical assistant robot will be deployed.

  1. Conclusion (new version)

In this work, we demonstrate the effectiveness of using a statistical approach to reduce errors in speech recognition systems. The Levenshtein and Damerau algorithms were compared by varying a threshold to minimize the WER on MoroccanFrench test set. The results showed that the Damerau algorithm was more effective than the Levenshtein algorithm in reducing the WER, with a threshold of max=0.8.

Our results are in line with other studies [1], [2] that have reported the effectiveness of post-processing methods in reducing errors for recognizing OOV words in ASR systems.

The post-processing method was used to detect and correct OOV words in the text after the speech recognition task. In terms of time consumption, the post-processing took only 0.029 seconds. In terms of WER reduction, it was between 1% and 4%, which indicates the method succeeded in reducing the error without modifying the ASR system itself. However, there is still room for improvement in this method.

In addition to the points mentioned previously, it is worth noting that the findings of this research have important theoretical and practical implications for the field of speech recognition. Theoretically, this research demonstrates the effectiveness of using statistical approaches to reduce errors in off-the-shelf speech recognition systems, which can inform future research in this area. Practically, the results of this research have potential applications in real-world scenarios such as speech-controlled medical assistant robots that can be implemented in retirement homes.

However, it is important to note that there are limitations to this research. One limitation is that the dataset used for evaluation has a relatively low vocabulary of possible misrecognized OOV words. Additionally, the testing dataset is noisy and relatively small, which may limit the generalizability of the results.

To address these limitations, we will expand the dataset to include more diverse accents and languages, and to explore the use of other post-processing algorithms and machine learning techniques to improve the accuracy of speech recognition systems.

References

  1. Réveil, K. Demuynck, and J. P. Martens, “An improved two-stage mixed language model approach for handling out-of-vocabulary words in large vocabulary continuous speech recognition,” Comput Speech Lang, vol. 28, no. 1, pp. 141–162, Jan. 2014, doi: 10.1016/J.CSL.2013.04.003. 
  2. ​A. Y. Andrusenko and A. N. Romanenko, “Improving out of vocabulary words recognition accuracy for an end-to-end Russian speech recognition system,” Scientific and Technical Journal of Information Technologies, Mechanics and Optics, vol. 22, no. 6, pp. 1143–1149, Nov. 2022, doi: 10.17586/2226-1494-2022-22-6-1143-1149. 

Reviewer 3 Report

Review of the Manuscript

"Adapting Off-the-shelf speech recognition systems for novel words"

The authors develop a post processing method to improve the correction of "out of vocabulary" word errors in speech recognition systems. The methodology uses a statistical approach based on distance. They use the Levenshtein distance and the Damerau Levenshtein distance. 

The problem under investigation is essential, and the methodology is interesting as it can be applied to any speech recognition system. The methodology is tested on a Moroccan French corpus using speech recognition systems trained on general French (this produces high error rates on the off the shelf systems).

The paper is short, engaging, and well written. The authors show that their procedure work on the corpus, the reduction of the error rates seems small, but this is a difficult problem. Also, the dataset is somewhat extreme, as shown by the error rates given by the off the shelf speech recognition systems. In my opinion, the methodology is promising and, without a doubt, will be improved.  

Author Response

Response to Reviewer 3 Comments

Point 1: 

"Adapting Off-the-shelf speech recognition systems for novel words"

The authors develop a post processing method to improve the correction of "out of vocabulary" word errors in speech recognition systems. The methodology uses a statistical approach based on distance. They use the Levenshtein distance and the Damerau Levenshtein distance. 

The problem under investigation is essential, and the methodology is interesting as it can be applied to any speech recognition system. The methodology is tested on a Moroccan French corpus using speech recognition systems trained on general French (this produces high error rates on the off the shelf systems).

The paper is short, engaging, and well written. The authors show that their procedure work on the corpus, the reduction of the error rates seems small, but this is a difficult problem. Also, the dataset is somewhat extreme, as shown by the error rates given by the off the shelf speech recognition systems. In my opinion, the methodology is promising and, without a doubt, will be improved.  

Response 1: 

Yes, thank you again, this is exactly what we have done in this study. In the conclusion, we mention the solutions that we will proceed with as improvements.

  1. Conclusion (new version)

In this work, we demonstrate the effectiveness of using a statistical approach to reduce errors in speech recognition systems. The Levenshtein and Damerau algorithms were compared by varying a threshold to minimize the WER on MoroccanFrench test set. The results showed that the Damerau algorithm was more effective than the Levenshtein algorithm in reducing the WER, with a threshold of max=0.8.

Our results are in line with other studies [1], [2] that have reported the effectiveness of post-processing methods in reducing errors for recognizing OOV words in ASR systems.

The post-processing method was used to detect and correct OOV words in the text after the speech recognition task. In terms of time consumption, the post-processing took only 0.029 seconds. In terms of WER reduction, it was between 1% and 4%, which indicates the method succeeded in reducing the error without modifying the ASR system itself. However, there is still room for improvement in this method.

In addition to the points mentioned previously, it is worth noting that the findings of this research have important theoretical and practical implications for the field of speech recognition. Theoretically, this research demonstrates the effectiveness of using statistical approaches to reduce errors in off-the-shelf speech recognition systems, which can inform future research in this area. Practically, the results of this research have potential applications in real-world scenarios such as speech-controlled medical assistant robots that can be implemented in retirement homes.

However, it is important to note that there are limitations to this research. One limitation is that the dataset used for evaluation has a relatively low vocabulary of possible misrecognized OOV words. Additionally, the testing dataset is noisy and relatively small, which may limit the generalizability of the results.

To address these limitations, we will expand the dataset to include more diverse accents and languages, and to explore the use of other post-processing algorithms and machine learning techniques to improve the accuracy of speech recognition systems.

Reviewer 4 Report

The introduction section is poor, only one reference is used. Please, elaborate the section thoroughly and rely more on literature.

Line 31. the authors claim »Unfortunately, most speech recognition systems are closed vocabulary and do not support OOV terms« Please, add reference.

Line 37: »Current OOV reseach focuses on…« Please, add references of the current research.

Althgouh in the related works section the authors provide a brief review of existing work, the research gap is not sufficiently presented and justified. Please, correct.

Fig 1 is difficult to read. Please, correct the small text.

The presentation of results is very basic. Could the authors more throroughly discuss the results?

The conclusion section is very poor. Please, extend it. It is crucial to add more relevant literature to the manuscript – the latest journal papers!! And, then, please, discuss your findings with the existing ones. In the conclusion, please, add also theoretical and practical implications, and limitations, and future outlook.

Author Response

Response to Reviewer 4 Comments

Point 1:
The introduction section is poor, only one reference is used. Please, elaborate the section thoroughly and rely more on literature.

Line 31. the authors claim »Unfortunately, most speech recognition systems are closed vocabulary and do not support OOV terms« Please, add reference.

Line 37: »Current OOV research focuses on…« Please, add references of the current research.

Response 1: 

For the introduction we included additional references to support and confirm our claims

Section : 2. Introduction (new version)

The development of assistant robots that possess the capability of natural communication and the ability to exhibit empathy towards individuals, similar to that of a human, would represent a significant breakthrough in the field of robotics.

The robot assistant is primarily used as a voice assistant with a physical body [1]​. A voice assistant is a system that uses a speech recognition module, a speech synthesis module, and a natural language processing (NLP) module to communicate like a human [2].

Ensuring effective speech recognition is crucial for the proper functioning of robot assistants. Non-native accents, new vocabulary, and aging voices can cause malfunctioning of the speech recognition system. If this task is not executed correctly, the voice assistant will inevitably produce false or random responses.

One essential subject of speech recognition systems is Out-of-Vocabulary words (OOV) ​[3]​. OOV words are new terms that show up in the speech test set but do not exist in the recognition vocabulary. They are generally essential content terms like names and localities that include information that is critical to the performance of several speech recognition tasks. Unfortunately, most speech recognition systems are closed vocabulary and do not support OOV terms ​[4]​​[5], [6]​. When a speech recognition system encounters a new term, it may mistake it for a similar sounding word in its vocabulary, leading to errors in recognizing surrounding words. To address this, there is a significant interest in developing speech recognition systems that can accurately detect and correct out-of-vocabulary words.

Current OOV research has focused on detecting the presence of OOV words in test data​ [7]​ ​[8]​ . Solutions to this issue can be broadly categorized into two categories: detection-based approaches and correction-based approaches. The objective of detection-based approaches is to determine whether an error has occurred in the transcription by utilizing features derived from the automatic speech recognition (ASR) system, such as confidence scores, language model, and confusion network density ​[9]​. Correction-based approaches, on the other hand, rely on the results of error detection to replace any misrecognized words with the correct ones.

Following the comparison of the most effective French speech recognition systems with the lowest error rate in a realistic setting ​[10]​, it has been observed that many of these systems have difficulty recognizing speech within specific domains such as the medical field and Moroccan culture. As a result, we aim to investigate the current state of the art and devise ways to enhance the systems' capability to understand out-of-vocabulary words.

This paper is structured as follows. In Section 2, we review previous studies on enhancing speech recognition accuracy, including both statistical and deep learning approaches. Section 3 details the methodology employed in this study, including the corpus we constructed and its usage for testing purposes, a comparison of commonly used ASR systems, and post-processing method for improving speech recognition for novel words. The results of our research are discussed in Section 4. Finally, in the conclusion, we summarize our findings and outline potential directions for future research.

Point 2 :

Although in the related works section the authors provide a brief review of existing work, the research gap is not sufficiently presented and justified. Please, correct.

Response 2: 

For related works section, we included additional references for both statistical and deep learning approaches, and we explained why we chose to use the statistical approach.

  1. Related works on improving speech recognition accuracy

Statistical approaches   

These models use traditional statistical techniques such as n-grams, Hidden Markov Models (HMM), and specific linguistic rules to learn the probability distribution of words. They have been used in speech recognition systems for a long time, and they have been found to be effective in many cases, especially when dealing with low-resource languages.

Kunisetti [11] tried various approaches to improve the speech recognition accuracy for Telugu text. The approaches are 1) calculating distance using the Levenshtein distance algorithm and adding minimum distance variants to the static dictionary, 2) addition of the frequently occurring errors, 3) addition of variants in the language model, 4) changing the probability, and 5) transcription modification. All the approaches in which the author succeeded in a lower error rate when new variants were added to the dictionary.

In [12], the authors investigate the possibility of using sub-word vocabularies where words are split into frequent and common parts. Sub-word vocabularies are extracted by performing word decomposition on the text corpus and taking the 100 thousand most frequent units. For comparison, the full vocabulary of this corpus contains approximately 1.5 million surface forms. The authors explored two different methods: (1) fully unsupervised and data-driven word decomposition using the Morfessor tool and (2) word decomposition using a stemmer. The results of the experiment (2) show that this allows for a significantly reduce OOV rate.

In [13], the authors presented the implementation of rule-based grapheme-to-phoneme G2P to assist the automatic generation of OOV word pronunciation for a speech recognition system. The proposed system gained an 87.29% overall phoneme error rate. Meanwhile, the achievable phoneme error rate for Indonesian words is 93.65%.

Pellegrini and his team examined the use of a Markov Chain classifier with two states, error and correct, to model errors using a set of 15 common features in error detection. The system was tested on an American English broadcast news speech NIST corpus and accomplished 860 errors correctly detected with only a 16.7% classification error rate.

The paper ​[14]​ presents a two-stage technique for handling OOV words in speech recognition, using a mixed word/subword language model to transcribe OOV words into subword units in the first stage, and a look-up table to convert them to word hypotheses in the second stage. The technique was tested on out-of-domain data, and it was found to reduce the word error rate by 12% relative to a baseline system.

 This paper ​[15]​ presents a statistical approach for handling OOV words in large vocabulary continuous speech recognition by using a lexicon-free technique. The authors propose a method to generate a transcription for OOV words based on the subword unit hypothesis, which is generated by a language model trained on a large amount of text data. The method was found to be effective for OOV word recognition.

Deep Learning approach

On the other hand, deep learning approaches, such as neural networks, are based on the idea of learning representations of the input data. These models require large amounts of labeled data to train and learn the underlying structure of the speech input. But once trained, they are able to extract high-level features from the speech signals, such as phonemes and subwords, which makes them well suited to handling OOV words.

Sarma et al. [16] build an ASR errors detector and corrector using co-occurrence analysis. They introduced a novel unsupervised approach for detecting and correcting miss-recognized query words in a document collection. According to the authors, this method can produce high-precision targeted detection and correction of OOV words.

In the same context, Bassil and Semaan [17] proposed a post-editing ASR error correction method based on the Microsoft N-Gram dataset for detecting and correcting spelling errors generated by ASR systems. The detection process consists of detecting OOV word spelling errors in the Microsoft N-Gram dataset, and the correction process consists of two steps: the first one consists of generating correction suggestions for the detected word errors, and the second one, comprises a context-sensitive errors correction algorithm for selecting the best candidate for the correction. The error rate using the proposed method was around 2.4% on a dataset composed of a collection of five different English articles each with around 100 words read by five different speakers.

More recently, [18] proposes a text processing model after Chinese Speech Recognition. It combines bidirectional long short-term memory (Bi-LSTM) network and conditional random field in two stages: text error detection and text error correction respectively. Through verification and system test on the SIGHAN 2013 Chinese Spelling Check (CSC) dataset, the experimental outcomes indicate that the model can successfully enhance text following speech recognition accuracy.

In [19], the authors propose a machine-translation inspired sequence-to-sequence approach which learns to “translate” hypotheses to reference transcripts. To augment training data, authors use all N-best hypotheses to form pairs with reference sentences, generate audio data using speech synthesis and add noise to the source recordings. The resulting training set consists of 640 M reference hypothesis pairs. The proposed system achieves an 18.6% relative WER (Word Error Rate) reduction.

Another paper [20] uses a similar approach for Mandarin speech recognition but proposes a Transformer model for spelling correction. The authors report a result of 22.9% relative CER (Character Error Rate) improvement.

In this paper [21], researchers present a syllabification algorithm and an end-to-end architecture for Amharic ASR that makes use of the language's phoneme-based subword units. According to the findings of the experiments, the speech recognition systems that use phoneme-based subwords that consider the context of the sentence are more accurate than the systems that use character-based, phoneme-based, and character-based subwords.

The paper ​[22]​ presents a method for handling OOV words in ASR systems that retrains an integral ASR system with a discriminative loss function and a decoding method based on a TG graph, using an open data set of the Russian language. The method aims to reduce the WER while maintaining the ability to recognize OOV words. Results show that it reduces the WER by 3% compared to the standard method, making the system more resistant to recognizing new unseen words.

The paper ​[23]​ presents FastContext, a method for handling out-of-vocabulary (OOV) words in natural language processing systems. It improves the embedding of subword information using a context-based embedding computed by a deep learning model. The method was evaluated on tasks of word similarity, named entity recognition, and part-of-speech tagging and performed better than FastText and other state-of-the-art OOV handling techniques. The results indicate that the approach is able to capture semantic or morphological information and it's more effective when the context is the most relevant source to infer the meaning of the OOV words.

The paper ​[24]​ describes the THUEE team's approach for handling Out-Of-Vocabulary (OOV) words in their speech recognition system for the IARPA Open Automatic Speech Recognition Challenge (OpenASR21). They used Grapheme-to-Phoneme (G2P) techniques to extend the pronunciation lexicon for OOV and potential new words in the Constrained training condition. They also applied multiple data augmentations techniques. In the Constrained-plus training condition, they used the self-supervised learning framework wav2vec2.0 and experimented with various fine-tuning techniques with the Connectionist Temporal Classification (CTC) criterion on top of the publicly available pre-trained model XLSR-53. They found that using the CTC model fine-tuned in the target language as the frontend feature extractor was effective in improving the OOV handling performance.

Since deep learning methods require more data, we will start with an effective statistical approach. statistical techniques can be better than modern techniques for handling OOV words in speech recognition for low-resource languages because they require less data to train, are more interpretable, and can be a good fit for languages with simple grammars and orthographic structures.

Point 3:

Fig 1 is difficult to read. Please, correct the small text.

Response 3: 

We attempted to improve the clarity of the figure.

Point 4:

The presentation of results is very basic. Could the authors more thoroughly discuss the results?

Response 4:  results and discussion (new version)

The table 4 shows the results of evaluating different automatic speech recognition (ASR) systems on MoroccanFrench test set, both before and after post-processing. The table compares the word error rate (WER) of each ASR system before and after post-processing, and also show a comparison between two different algorithms of post-processing, Levenshtein and Damerau Levenshtein. WER is a measure of the accuracy of the transcription, with a lower WER indicating better performance.

The results in the table indicate that the WER of the ASR systems is high before post-processing. For example, the WER of the Speech-to-text system is 38.27%, the Wav2vec 2.0 system is 55.96%, and the Quartznet system is 48.35%. This is due to the specific dataset used for evaluation, which is designed to simulate real-world scenarios with noise and different accents and new vocabulary, making the transcription more difficult.

After post-processing, the WER of the ASR systems is reduced, indicating that post-processing improves the accuracy of the transcription. For example, the WER of the Speech-to-text system is reduced to 37%, the Wav2vec 2.0 system is reduced to 51.71%, and the Quartznet system is reduced to 48.04%.

The post-processing method used in this evaluation is based on the Levenshtein and Damerau Levenshtein algorithm. The threshold for the Levenshtein algorithm is set to the minimum, where a threshold of 0.2 means that two words are similar. On the other hand, for the Damerau algorithm, the threshold is set to the maximum, where a threshold of 0.8, 0.85 or 0.9 means that two words are similar.

Wav2vec 2.0 was able to significantly reduce the error rate due to its use of an acoustic model without a language model, which resulted in the generation of words that were phonetically correct but had incorrect orthography. This made our approach more effective. Our tests indicate that speech recognition systems that are paired with a language model trained in various domains may produce lexically correct but contextually incorrect predictions (as seen in Table 5). For instance, a simple sentence predicted by a speech-to-text system that has not been trained on the word "mascir" may make predictions based on acoustic similarities and identify it as a phonetically matching in-vocabulary item, such as in the case of "mascir" and "ma soeur".

  • Mascir relève de l’unsiversité Mohammed VI polytechnique.
  • Ma sœur relève de l’université Mohammed VI polytechnique.

This prompted us to consider incorporating a language model that comprehends the context of the sentence for a more accurate result.

Point 5:
The conclusion section is very poor. Please, extend it. It is crucial to add more relevant literature to the manuscript – the latest journal papers!! And, then, please, discuss your findings with the existing ones. In the conclusion, please, add also theoretical and practical implications, and limitations, and future outlook.

Response 5:

As we were the first ones to try combining distance algorithms with commonly used speech recognition systems such as Google Speech-to-text, Vosk, SpeechBrain, QuartzNet, and Wav2Vec, we were unable to compare our results to other studies as we used our own test set for evaluation. Nevertheless, our findings are consistent with those of other research that have similarities to our approach.

Section: 5. Conclusion (new version)

 In this work, we demonstrate the effectiveness of using a statistical approach to reduce errors in speech recognition systems. The Levenshtein and Damerau algorithms were compared by varying a threshold to minimize the WER on MoroccanFrench test set. The results showed that the Damerau algorithm was more effective than the Levenshtein algorithm in reducing the WER, with a threshold of max=0.8.

Our results are in line with other studies [14], [22] that have reported the effective-ness of post-processing methods in reducing errors for recognizing OOV words in ASR systems.

The post-processing method was used to detect and correct OOV words in the text after the speech recognition task. In terms of time consumption, the post-processing took only 0.029 seconds. In terms of WER reduction, it was between 1% and 4%, which indicates the method succeeded in reducing the error without modifying the ASR system itself. However, there is still room for improvement in this method.

In addition to the points mentioned previously, it is worth noting that the findings of this research have important theoretical and practical implications for the field of speech recognition. Theoretically, this research demonstrates the effectiveness of using statistical approaches to reduce errors in off-the-shelf speech recognition systems, which can inform future research in this area. Practically, the results of this research have potential applications in real-world scenarios such as speech-controlled medical assistant robots that can be implemented in retirement homes.

However, it is important to note that there are limitations to this research. One limitation is that the dataset used for evaluation has a relatively low vocabulary of possible misrecognized OOV words. Additionally, the testing dataset is noisy and relatively small, which may limit the generalizability of the results.

To address these limitations, we will expand the dataset to include more diverse accents and languages, and to explore the use of other post-processing algorithms and machine learning techniques to improve the accuracy of speech recognition systems.

 References

  1. Pollmann, C. Ruff, K. Vetter, and G. Zimmermann, “Robot vs. voice assistant: Is playing with pepper more fun than playing with alexa?,” ACM/IEEE International Conference on Human-Robot Interaction, pp. 395–397, Mar. 2020, doi: 10.1145/3371382.3378251. 
  2. ​“How to Build Domain Specific Automatic Speech Recognition Models on GPUs | NVIDIA Developer Blog.” https://developer.nvidia.com/blog/how-to-build-domain-specific-automatic-speech-recognition-models-on-gpus/ (accessed Nov. 29, 2021). 
  3. ​T. Desot, F. Portet, and M. Vacher, “End-to-End Spoken Language Understanding: Performance analyses of a voice command task in a low resource setting,” Comput Speech Lang, vol. 75, p. 101369, Sep. 2022, doi: 10.1016/J.CSL.2022.101369. 
  4. ​J. Kim and P. Kang, “K-Wav2vec 2.0: Automatic Speech Recognition based on Joint Decoding of Graphemes and Syllables,” arXiv:2110.05172v1, 2021. 
  5. ​A. Laptev, A. Andrusenko, I. Podluzhny, A. Mitrofanov, I. Medennikov, and Y. Matveev, “Dynamic Acoustic Unit Augmentation with BPE-Dropout for Low-Resource End-to-End Speech Recognition,” Sensors 2021, Vol. 21, Page 3063, vol. 21, no. 9, p. 3063, Apr. 2021, doi: 10.3390/S21093063. 
  6. ​A. Y. Andrusenko and A. N. Romanenko, “Improving out of vocabulary words recognition accuracy for an end-to-end Russian speech recognition system,” Scientific and Technical Journal of Information Technologies, Mechanics and Optics, vol. 22, no. 6, pp. 1143–1149, Nov. 2022, doi: 10.17586/2226-1494-2022-22-6-1143-1149. 
  7. ​J. v. Lochter, R. M. Silva, and T. A. Almeida, “Multi-level out-of-vocabulary words handling approach,” Knowl Based Syst, vol. 251, Sep. 2022, doi: 10.1016/J.KNOSYS.2022.108911. 
  8. ​F. Y. Putri, D. Hoesen, and D. P. Lestari, “Rule-Based Pronunciation Models to Handle OOV Words for Indonesian Automatic Speech Recognition System,” Proceeding - 2019 5th International Conference on Science in Information Technology: Embracing Industry 4.0: Towards Innovation in Cyber Physical System, ICSITech 2019, pp. 246–251, Oct. 2019, doi: 10.1109/ICSITECH46713.2019.8987472.
  9. Errattahi, A. El Hannani, and H. Ouahmane, “Automatic speech recognition errors detection and correction: A review,” Procedia Computer Science, vol. 128, pp. 32–37, 2018, doi: 10.1016/J.PROCS.2018.03.005.
  10. ​ Fadel, I. Araf, T. Bouchentouf, P.-A. Buvet, F. Bourzeix, and O. Bourja, “Which French speech recognition system for assistant robots?,” pp. 1–5, Mar. 2022, doi: 10.1109/IRASET52964.2022.9737976. 
  11. Subramanyam, “Improving Speech Recognition Accuracy Using Levenshtein Distance algorithm”.
  12. Salimbajevs, “Using sub-word n-gram models for dealing with OOV in large vocabulary speech recognition for Latvian”.
  13. Y. Putri, D. Hoesen, and D. P. Lestari, “Rule-Based Pronunciation Models to Handle OOV Words for Indonesian Automatic Speech Recognition System,” Proceeding - 2019 5th International Conference on Science in Information Technology: Embracing Industry 4.0: Towards Innovation in Cyber Physical System, ICSITech 2019, pp. 246–251, Oct. 2019, doi: 10.1109/ICSITECH46713.2019.8987472.
  14. Réveil, K. Demuynck, and J. P. Martens, “An improved two-stage mixed language model approach for handling out-of-vocabulary words in large vocabulary continuous speech recognition,” Comput Speech Lang, vol. 28, no. 1, pp. 141–162, Jan. 2014, doi: 10.1016/J.CSL.2013.04.003. 
  15. ​[12] Pellegrini and I. Trancoso, “Error detection in broadcast news ASR using Markov chains,” Lecture Notes in Computer Science (including subseries Lecture Notes in Artificial Intelligence and Lecture Notes in Bioinformatics), vol. 6562 LNAI, pp. 59–69, 2011, doi: 10.1007/978-3-642-20095-3_6/COVER. 
  16. Sarma and D. D. Palmer, “Context-based Speech Recognition Error Detection and Correction”.
  17. Bassil and P. Semaan, “ASR Context-Sensitive Error Correction Based on Microsoft N-Gram Dataset,” Mar. 2012, doi: 10.48550/arxiv.1203.5262.
  18. Yang, Y. Li, J. Wang, and Z. Tang, “Post Text Processing of Chinese Speech Recognition Based on Bidirectional LSTM Networks and CRF,” Electronics 2019, Vol. 8, Page 1248, vol. 8, no. 11, p. 1248, Oct. 2019, doi: 10.3390/ELECTRONICS8111248.
  19. “(2) (PDF) Post-editing and Rescoring of ASR Results with Edit Operations Tagging.” https://www.researchgate.net/publication/351687519_Post-editing_and_Rescoring_of_ASR_Results_with_Edit_Operations_Tagging (accessed Jul. 25, 2022).
  20. Zhang, M. Lei, and Z. Yan, “Automatic Spelling Correction with Transformer for CTC-based End-to-End Speech Recognition,” Mar. 2019, doi: 10.48550/arxiv.1904.10045.
  21. D. Emiru, S. Xiong, Y. Li, A. Fesseha, and M. Diallo, “Improving amharic speech recognition system using connectionist temporal classification with attention model and phoneme-based byte-pair-encodings,” Information (Switzerland), vol. 12, no. 2, pp. 1–22, Feb. 2021, doi: 10.3390/INFO12020062.
  22. Y. Andrusenko and A. N. Romanenko, “Improving out of vocabulary words recognition accuracy for an end-to-end Russian speech recognition system,” Scientific and Technical Journal of Information Technologies, Mechanics and Optics, vol. 22, no. 6, pp. 1143–1149, Nov. 2022, doi: 10.17586/2226-1494-2022-22-6-1143-1149. 
  23. ​R. M. Silva, J. v. Lochter, T. A. Almeida, and A. Yamakami, “FastContext: Handling Out-of-Vocabulary Words Using the Word Structure and Context,” Lecture Notes in Computer Science (including subseries Lecture Notes in Artificial Intelligence and Lecture Notes in Bioinformatics), vol. 13654 LNAI, pp. 539–557, 2022, doi: 10.1007/978-3-031-21689-3_38. 
  24. ​J. Zhao et al., “The THUEE System Description for the IARPA OpenASR21 Challenge,” Proceedings of the Annual Conference of the International Speech Communication Association, INTERSPEECH, vol. 2022-September, pp. 4855–4859, 2022, doi: 10.21437/INTERSPEECH.2022-269. 
  25. L. Aouragh, A. Yousfi, U. Sidi, M. Ben Abdellah, and G. Hicham, “Adapting the Levenshtein Distance to Contextual Spelling Correction,” International Journal of Computer Science and Applications, ÓTechnomathematics Research Foundation, vol. 12, no. 1, pp. 127–133, 2015.

Reviewer 5 Report

The paper is discussing an emerging topic in NLP which is speech recognition.

The quality of paper is good but needs some modifications:

1) the literature is outdated. Some new references from 2020-2022 needs to be added.

2) compare the results with new articles from recent literature.

3) to increase the ineterest to the reader, it is important to highlight the applications of this work such as in chatbots and cheating detection. Here are some recent references that the author can refer to:

https://www.mdpi.com/1424-8220/22/14/5381

https://www.mdpi.com/2071-1050/14/3/1777

https://www.sciencedirect.com/science/article/pii/S2667305322000904

https://www.mdpi.com/1424-8220/22/3/1228

Author Response

Response to Reviewer 5 Comments

Point 1: 

The paper is discussing an emerging topic in NLP which is speech recognition.

The quality of paper is good but needs some modifications:

  • the literature is outdated. Some new references from 2020-2022 needs to be added.

Response 1: 

Thank you, for related works section, we included additional recent references for both statistical and deep learning approaches, and we explained why we chose to use the statistical approach.

  1. Related works on improving speech recognition accuracy

Statistical approaches   

These models use traditional statistical techniques such as n-grams, Hidden Markov Models (HMM), and specific linguistic rules to learn the probability distribution of words. They have been used in speech recognition systems for a long time, and they have been found to be effective in many cases, especially when dealing with low-resource languages.

Kunisetti [11] tried various approaches to improve the speech recognition accuracy for Telugu text. The approaches are 1) calculating distance using the Levenshtein distance algorithm and adding minimum distance variants to the static dictionary, 2) addition of the frequently occurring errors, 3) addition of variants in the language model, 4) changing the probability, and 5) transcription modification. All the approaches in which the author succeeded in a lower error rate when new variants were added to the dictionary.

In [12], the authors investigate the possibility of using sub-word vocabularies where words are split into frequent and common parts. Sub-word vocabularies are extracted by performing word decomposition on the text corpus and taking the 100 thousand most frequent units. For comparison, the full vocabulary of this corpus contains approximately 1.5 million surface forms. The authors explored two different methods: (1) fully unsupervised and data-driven word decomposition using the Morfessor tool and (2) word decomposition using a stemmer. The results of the experiment (2) show that this allows for a significantly reduce OOV rate.

In [13], the authors presented the implementation of rule-based grapheme-to-phoneme G2P to assist the automatic generation of OOV word pronunciation for a speech recognition system. The proposed system gained an 87.29% overall phoneme error rate. Meanwhile, the achievable phoneme error rate for Indonesian words is 93.65%.

Pellegrini and his team examined the use of a Markov Chain classifier with two states, error and correct, to model errors using a set of 15 common features in error detection. The system was tested on an American English broadcast news speech NIST corpus and accomplished 860 errors correctly detected with only a 16.7% classification error rate.

The paper ​[14]​ presents a two-stage technique for handling OOV words in speech recognition, using a mixed word/subword language model to transcribe OOV words into subword units in the first stage, and a look-up table to convert them to word hypotheses in the second stage. The technique was tested on out-of-domain data, and it was found to reduce the word error rate by 12% relative to a baseline system.

 This paper ​[15]​ presents a statistical approach for handling OOV words in large vocabulary continuous speech recognition by using a lexicon-free technique. The authors propose a method to generate a transcription for OOV words based on the subword unit hypothesis, which is generated by a language model trained on a large amount of text data. The method was found to be effective for OOV word recognition.

Deep Learning approach

On the other hand, deep learning approaches, such as neural networks, are based on the idea of learning representations of the input data. These models require large amounts of labeled data to train and learn the underlying structure of the speech input. But once trained, they are able to extract high-level features from the speech signals, such as phonemes and subwords, which makes them well suited to handling OOV words.

Sarma et al. [16] build an ASR errors detector and corrector using co-occurrence analysis. They introduced a novel unsupervised approach for detecting and correcting miss-recognized query words in a document collection. According to the authors, this method can produce high-precision targeted detection and correction of OOV words.

In the same context, Bassil and Semaan [17] proposed a post-editing ASR error correction method based on the Microsoft N-Gram dataset for detecting and correcting spelling errors generated by ASR systems. The detection process consists of detecting OOV word spelling errors in the Microsoft N-Gram dataset, and the correction process consists of two steps: the first one consists of generating correction suggestions for the detected word errors, and the second one, comprises a context-sensitive errors correction algorithm for selecting the best candidate for the correction. The error rate using the proposed method was around 2.4% on a dataset composed of a collection of five different English articles each with around 100 words read by five different speakers.

More recently, [18] proposes a text processing model after Chinese Speech Recognition. It combines bidirectional long short-term memory (Bi-LSTM) network and conditional random field in two stages: text error detection and text error correction respectively. Through verification and system test on the SIGHAN 2013 Chinese Spelling Check (CSC) dataset, the experimental outcomes indicate that the model can successfully enhance text following speech recognition accuracy.

In [19], the authors propose a machine-translation inspired sequence-to-sequence approach which learns to “translate” hypotheses to reference transcripts. To augment training data, authors use all N-best hypotheses to form pairs with reference sentences, generate audio data using speech synthesis and add noise to the source recordings. The resulting training set consists of 640 M reference hypothesis pairs. The proposed system achieves an 18.6% relative WER (Word Error Rate) reduction.

Another paper [20] uses a similar approach for Mandarin speech recognition but proposes a Transformer model for spelling correction. The authors report a result of 22.9% relative CER (Character Error Rate) improvement.

In this paper [21], researchers present a syllabification algorithm and an end-to-end architecture for Amharic ASR that makes use of the language's phoneme-based subword units. According to the findings of the experiments, the speech recognition systems that use phoneme-based subwords that consider the context of the sentence are more accurate than the systems that use character-based, phoneme-based, and character-based subwords.

The paper ​[22]​ presents a method for handling OOV words in ASR systems that retrains an integral ASR system with a discriminative loss function and a decoding method based on a TG graph, using an open data set of the Russian language. The method aims to reduce the WER while maintaining the ability to recognize OOV words. Results show that it reduces the WER by 3% compared to the standard method, making the system more resistant to recognizing new unseen words.

The paper ​[23]​ presents FastContext, a method for handling out-of-vocabulary (OOV) words in natural language processing systems. It improves the embedding of subword information using a context-based embedding computed by a deep learning model. The method was evaluated on tasks of word similarity, named entity recognition, and part-of-speech tagging and performed better than FastText and other state-of-the-art OOV handling techniques. The results indicate that the approach is able to capture semantic or morphological information and it's more effective when the context is the most relevant source to infer the meaning of the OOV words.

The paper ​[24]​ describes the THUEE team's approach for handling Out-Of-Vocabulary (OOV) words in their speech recognition system for the IARPA Open Automatic Speech Recognition Challenge (OpenASR21). They used Grapheme-to-Phoneme (G2P) techniques to extend the pronunciation lexicon for OOV and potential new words in the Constrained training condition. They also applied multiple data augmentations techniques. In the Constrained-plus training condition, they used the self-supervised learning framework wav2vec2.0 and experimented with various fine-tuning techniques with the Connectionist Temporal Classification (CTC) criterion on top of the publicly available pre-trained model XLSR-53. They found that using the CTC model fine-tuned in the target language as the frontend feature extractor was effective in improving the OOV handling performance.

Since deep learning methods require more data, we will start with an effective statistical approach. statistical techniques can be better than modern techniques for handling OOV words in speech recognition for low-resource languages because they require less data to train, are more interpretable, and can be a good fit for languages with simple grammars and orthographic structures.

Point 2: 

2) compare the results with new articles from recent literature.

Response 2 :

As we were the first ones to try combining distance algorithms with commonly used speech recognition systems such as Google Speech-to-text, Vosk, SpeechBrain, QuartzNet, and Wav2Vec, we were unable to compare our results to other studies as we used our own test set for evaluation. Nevertheless, our findings are consistent with those of other research that have similarities to our approach.

Section: 5. Conclusion

 In this work, we demonstrate the effectiveness of using a statistical approach to reduce errors in speech recognition systems. The Levenshtein and Damerau algorithms were compared by varying a threshold to minimize the WER on MoroccanFrench test set. The results showed that the Damerau algorithm was more effective than the Levenshtein algorithm in reducing the WER, with a threshold of max=0.8.

Our results are in line with other studies [14], [22] that have reported the effective-ness of post-processing methods in reducing errors for recognizing OOV words in ASR systems.

The post-processing method was used to detect and correct OOV words in the text after the speech recognition task. In terms of time consumption, the post-processing took only 0.029 seconds. In terms of WER reduction, it was between 1% and 4%, which indicates the method succeeded in reducing the error without modifying the ASR system itself. However, there is still room for improvement in this method.

In addition to the points mentioned previously, it is worth noting that the findings of this research have important theoretical and practical implications for the field of speech recognition. Theoretically, this research demonstrates the effectiveness of using statistical approaches to reduce errors in off-the-shelf speech recognition systems, which can inform future research in this area. Practically, the results of this research have potential applications in real-world scenarios such as speech-controlled medical assistant robots that can be implemented in retirement homes.

However, it is important to note that there are limitations to this research. One limitation is that the dataset used for evaluation has a relatively low vocabulary of pos-sible misrecognized OOV words. Additionally, the testing dataset is noisy and relative-ly small, which may limit the generalizability of the results.

To address these limitations, we will expand the dataset to include more diverse accents and languages, and to explore the use of other post-processing algorithms and machine learning techniques to improve the accuracy of speech recognition systems.

Point 3:

3) to increase the interest to the reader, it is important to highlight the applications of this work such as in chatbots and cheating detection. Here are some recent references that the author can refer to:

https://www.mdpi.com/1424-8220/22/14/5381

https://www.mdpi.com/2071-1050/14/3/1777

https://www.sciencedirect.com/science/article/pii/S2667305322000904

https://www.mdpi.com/1424-8220/22/3/1228

Response 3:

We discuss the applications and implications of the research in the introduction and conclusion sections.

Section: 2. Introduction (new version)

The development of assistant robots that possess the capability of natural communication and the ability to exhibit empathy towards individuals, similar to that of a human, would represent a significant breakthrough in the field of robotics.

The robot assistant is primarily used as a voice assistant with a physical body [1]​. A voice assistant is a system that uses a speech recognition module, a speech synthesis module, and a natural language processing (NLP) module to communicate like a human [2].

Ensuring effective speech recognition is crucial for the proper functioning of robot assistants. Non-native accents, new vocabulary, and aging voices can cause malfunctioning of the speech recognition system. If this task is not executed correctly, the voice assistant will inevitably produce false or random responses.

One essential subject of speech recognition systems is Out-of-Vocabulary words (OOV) ​[3]​. OOV words are new terms that show up in the speech test set but do not exist in the recognition vocabulary. They are generally essential content terms like names and localities that include information that is critical to the performance of several speech recognition tasks. Unfortunately, most speech recognition systems are closed vocabulary and do not support OOV terms ​[4]​​[5], [6]​. When a speech recognition system encounters a new term, it may mistake it for a similar sounding word in its vocabulary, leading to errors in recognizing surrounding words. To address this, there is a significant interest in developing speech recognition systems that can accurately detect and correct out-of-vocabulary words.

Current OOV research has focused on detecting the presence of OOV words in test data​ [7]​ ​[8]​ . Solutions to this issue can be broadly categorized into two categories: detection-based approaches and correction-based approaches. The objective of detection-based approaches is to determine whether an error has occurred in the transcription by utilizing features derived from the automatic speech recognition (ASR) system, such as confidence scores, language model, and confusion network density ​[9]​. Correction-based approaches, on the other hand, rely on the results of error detection to replace any misrecognized words with the correct ones.

Following the comparison of the most effective French speech recognition systems with the lowest error rate in a realistic setting ​[10]​, it has been observed that many of these systems have difficulty recognizing speech within specific domains such as the medical field and Moroccan culture. As a result, we aim to investigate the current state of the art and devise ways to enhance the systems' capability to understand out-of-vocabulary words.

This paper is structured as follows. In Section 2, we review previous studies on enhancing speech recognition accuracy, including both statistical and deep learning approaches. Section 3 details the methodology employed in this study, including the corpus we constructed and its usage for testing purposes, a comparison of commonly used ASR systems, and post-processing method for improving speech recognition for novel words. The results of our research are discussed in Section 4. Finally, in the conclusion, we summarize our findings and outline potential directions for future research.

Section: 5. Conclusion (new version)

In this work, we demonstrate the effectiveness of using a statistical approach to reduce errors in speech recognition systems. The Levenshtein and Damerau algorithms were compared by varying a threshold to minimize the WER on MoroccanFrench test set. The results showed that the Damerau algorithm was more effective than the Levenshtein algorithm in reducing the WER, with a threshold of max=0.8.

Our results are in line with other studies[14], [22] that have reported the effectiveness of post-processing methods in reducing errors for recognizing OOV words in ASR systems.

The post-processing method was used to detect and correct OOV words in the text after the speech recognition task. In terms of time consumption, the post-processing took only 0.029 seconds. In terms of WER reduction, it was between 1% and 4%, which indicates the method succeeded in reducing the error without modifying the ASR system itself. However, there is still room for improvement in this method.

In addition to the points mentioned previously, it is worth noting that the findings of this research have important theoretical and practical implications for the field of speech recognition. Theoretically, this research demonstrates the effectiveness of using statistical approaches to reduce errors in off-the-shelf speech recognition systems, which can inform future research in this area. Practically, the results of this research have potential applications in real-world scenarios such as speech-controlled medical assistant robots that can be implemented in retirement homes.

However, it is important to note that there are limitations to this research. One limitation is that the dataset used for evaluation has a relatively low vocabulary of possible misrecognized OOV words. Additionally, the testing dataset is noisy and relatively small, which may limit the generalizability of the results.

To address these limitations, we will expand the dataset to include more diverse accents and languages, and to explore the use of other post-processing algorithms and machine learning techniques to improve the accuracy of speech recognition systems.

References

  1. Pollmann, C. Ruff, K. Vetter, and G. Zimmermann, “Robot vs. voice assistant: Is playing with pepper more fun than playing with alexa?,” ACM/IEEE International Conference on Human-Robot Interaction, pp. 395–397, Mar. 2020, doi: 10.1145/3371382.3378251. 
  2. ​“How to Build Domain Specific Automatic Speech Recognition Models on GPUs | NVIDIA Developer Blog.” https://developer.nvidia.com/blog/how-to-build-domain-specific-automatic-speech-recognition-models-on-gpus/ (accessed Nov. 29, 2021). 
  3. ​T. Desot, F. Portet, and M. Vacher, “End-to-End Spoken Language Understanding: Performance analyses of a voice command task in a low resource setting,” Comput Speech Lang, vol. 75, p. 101369, Sep. 2022, doi: 10.1016/J.CSL.2022.101369. 
  4. ​J. Kim and P. Kang, “K-Wav2vec 2.0: Automatic Speech Recognition based on Joint Decoding of Graphemes and Syllables,” arXiv:2110.05172v1, 2021. 
  5. ​A. Laptev, A. Andrusenko, I. Podluzhny, A. Mitrofanov, I. Medennikov, and Y. Matveev, “Dynamic Acoustic Unit Augmentation with BPE-Dropout for Low-Resource End-to-End Speech Recognition,” Sensors 2021, Vol. 21, Page 3063, vol. 21, no. 9, p. 3063, Apr. 2021, doi: 10.3390/S21093063. 
  6. ​A. Y. Andrusenko and A. N. Romanenko, “Improving out of vocabulary words recognition accuracy for an end-to-end Russian speech recognition system,” Scientific and Technical Journal of Information Technologies, Mechanics and Optics, vol. 22, no. 6, pp. 1143–1149, Nov. 2022, doi: 10.17586/2226-1494-2022-22-6-1143-1149. 
  7. ​J. v. Lochter, R. M. Silva, and T. A. Almeida, “Multi-level out-of-vocabulary words handling approach,” Knowl Based Syst, vol. 251, Sep. 2022, doi: 10.1016/J.KNOSYS.2022.108911. 
  8. ​F. Y. Putri, D. Hoesen, and D. P. Lestari, “Rule-Based Pronunciation Models to Handle OOV Words for Indonesian Automatic Speech Recognition System,” Proceeding - 2019 5th International Conference on Science in Information Technology: Embracing Industry 4.0: Towards Innovation in Cyber Physical System, ICSITech 2019, pp. 246–251, Oct. 2019, doi: 10.1109/ICSITECH46713.2019.8987472.
  9. Errattahi, A. El Hannani, and H. Ouahmane, “Automatic speech recognition errors detection and correction: A review,” Procedia Computer Science, vol. 128, pp. 32–37, 2018, doi: 10.1016/J.PROCS.2018.03.005.
  10. ​ Fadel, I. Araf, T. Bouchentouf, P.-A. Buvet, F. Bourzeix, and O. Bourja, “Which French speech recognition system for assistant robots?,” pp. 1–5, Mar. 2022, doi: 10.1109/IRASET52964.2022.9737976. 
  11. Subramanyam, “Improving Speech Recognition Accuracy Using Levenshtein Distance algorithm”.
  12. Salimbajevs, “Using sub-word n-gram models for dealing with OOV in large vocabulary speech recognition for Latvian”.
  13. Y. Putri, D. Hoesen, and D. P. Lestari, “Rule-Based Pronunciation Models to Handle OOV Words for Indonesian Automatic Speech Recognition System,” Proceeding - 2019 5th International Conference on Science in Information Technology: Embracing Industry 4.0: Towards Innovation in Cyber Physical System, ICSITech 2019, pp. 246–251, Oct. 2019, doi: 10.1109/ICSITECH46713.2019.8987472.
  14. Réveil, K. Demuynck, and J. P. Martens, “An improved two-stage mixed language model approach for handling out-of-vocabulary words in large vocabulary continuous speech recognition,” Comput Speech Lang, vol. 28, no. 1, pp. 141–162, Jan. 2014, doi: 10.1016/J.CSL.2013.04.003. 
  15. ​[12] Pellegrini and I. Trancoso, “Error detection in broadcast news ASR using Markov chains,” Lecture Notes in Computer Science (including subseries Lecture Notes in Artificial Intelligence and Lecture Notes in Bioinformatics), vol. 6562 LNAI, pp. 59–69, 2011, doi: 10.1007/978-3-642-20095-3_6/COVER. 
  16. Sarma and D. D. Palmer, “Context-based Speech Recognition Error Detection and Correction”.
  17. Bassil and P. Semaan, “ASR Context-Sensitive Error Correction Based on Microsoft N-Gram Dataset,” Mar. 2012, doi: 10.48550/arxiv.1203.5262.
  18. Yang, Y. Li, J. Wang, and Z. Tang, “Post Text Processing of Chinese Speech Recognition Based on Bidirectional LSTM Networks and CRF,” Electronics 2019, Vol. 8, Page 1248, vol. 8, no. 11, p. 1248, Oct. 2019, doi: 10.3390/ELECTRONICS8111248.
  19. “(2) (PDF) Post-editing and Rescoring of ASR Results with Edit Operations Tagging.” https://www.researchgate.net/publication/351687519_Post-editing_and_Rescoring_of_ASR_Results_with_Edit_Operations_Tagging (accessed Jul. 25, 2022).
  20. Zhang, M. Lei, and Z. Yan, “Automatic Spelling Correction with Transformer for CTC-based End-to-End Speech Recognition,” Mar. 2019, doi: 10.48550/arxiv.1904.10045.
  21. D. Emiru, S. Xiong, Y. Li, A. Fesseha, and M. Diallo, “Improving amharic speech recognition system using connectionist temporal classification with attention model and phoneme-based byte-pair-encodings,” Information (Switzerland), vol. 12, no. 2, pp. 1–22, Feb. 2021, doi: 10.3390/INFO12020062.
  22. Y. Andrusenko and A. N. Romanenko, “Improving out of vocabulary words recognition accuracy for an end-to-end Russian speech recognition system,” Scientific and Technical Journal of Information Technologies, Mechanics and Optics, vol. 22, no. 6, pp. 1143–1149, Nov. 2022, doi: 10.17586/2226-1494-2022-22-6-1143-1149. 
  23. ​R. M. Silva, J. v. Lochter, T. A. Almeida, and A. Yamakami, “FastContext: Handling Out-of-Vocabulary Words Using the Word Structure and Context,” Lecture Notes in Computer Science (including subseries Lecture Notes in Artificial Intelligence and Lecture Notes in Bioinformatics), vol. 13654 LNAI, pp. 539–557, 2022, doi: 10.1007/978-3-031-21689-3_38. 
  24. ​J. Zhao et al., “The THUEE System Description for the IARPA OpenASR21 Challenge,” Proceedings of the Annual Conference of the International Speech Communication Association, INTERSPEECH, vol. 2022-September, pp. 4855–4859, 2022, doi: 10.21437/INTERSPEECH.2022-269. 
  25. L. Aouragh, A. Yousfi, U. Sidi, M. Ben Abdellah, and G. Hicham, “Adapting the Levenshtein Distance to Contextual Spelling Correction,” International Journal of Computer Science and Applications, ÓTechnomathematics Research Foundation, vol. 12, no. 1, pp. 127–133, 2015.

Reviewer 6 Report

1. “3.3.1 Damerau Levenshtein algorithm” should be changed as “3.3.2 Damerau Levenshtein algorithm”.

2. Perhaps, the author could mention the contributions of practical applications in abstract and conclusion sections.

3. Perhaps, the authors could discuss more about the comparison with some traditional methods to highlight the advantages and contributions of the proposed method.

4. The manuscript should be proof read, otherwise it is difficult to understand and read. Please re-check the grammar and spelling.

Author Response

Response to Reviewer 6 Comments

Point 1:

  1. “3.3.1 Damerau Levenshtein algorithm” should be changed as “3.3.2 Damerau Levenshtein algorithm”.

Response 1: 

Thank you, the change has been done.

Point 2:

  1. Perhaps, the author could mention the contributions of practical applications in abstract and conclusion sections.

Response 2 : 

We mentioned the applications and implications of the research in abstract, introduction and conclusion sections.

Section : Abstract (new version)

Current speech recognition systems with fixed vocabulary have difficulties recognizing out-of-vocabulary words (OOVs) like names and novel words, this leads to misunderstanding or even task failure in dialog systems. Ensuring effective speech recognition is crucial for the proper functioning of robot assistants. Non-native accents, new vocabulary, and aging voices can cause malfunctioning of the speech recognition system. If this task is not executed correctly, the assistant robot will inevitably produce false or random responses. In this paper, we used a statistical approach based on distance algorithms to improve OOV correction. We developed a post-processing algorithm to be combined with speech recognition model. In this sense, we compared two distance algorithms: Damerau Levenshtein and Levenshtein distance. We validated the performance of the two distance algorithms in conjunction with five off-the-shelf speech recognition models. Damerau, as compared to the Levenshtein algorithm, succeeded to minimize the Word Error Rate (WER) on the Moroccan French test set for the five speech recognition systems, namely vosk API, google API, Wav2vec2.0, SpeechBrain, and Quartznet pre-trained models. Our post-processing method works regardless of the architecture of the speech recognizer, and the results on our MoroccanFrench test set outperform the five off-the-shelf speech recognizer systems.

Section : 5. Conclusion (new version)

In this work, we demonstrate the effectiveness of using a statistical approach to reduce errors in speech recognition systems. The Levenshtein and Damerau algorithms were compared by varying a threshold to minimize the WER on MoroccanFrench test set. The results showed that the Damerau algorithm was more effective than the Levenshtein algorithm in reducing the WER, with a threshold of max=0.8.

Our results are in line with other studies[14], [22] that have reported the effectiveness of post-processing methods in reducing errors for recognizing OOV words in ASR systems.

The post-processing method was used to detect and correct OOV words in the text after the speech recognition task. In terms of time consumption, the post-processing took only 0.029 seconds. In terms of WER reduction, it was between 1% and 4%, which indicates the method succeeded in reducing the error without modifying the ASR system itself. However, there is still room for improvement in this method.

In addition to the points mentioned previously, it is worth noting that the findings of this research have important theoretical and practical implications for the field of speech recognition. Theoretically, this research demonstrates the effectiveness of using statistical approaches to reduce errors in off-the-shelf speech recognition systems, which can inform future research in this area. Practically, the results of this research have potential applications in real-world scenarios such as speech-controlled medical assistant robots that can be implemented in retirement homes.

However, it is important to note that there are limitations to this research. One limitation is that the dataset used for evaluation has a relatively low vocabulary of possible misrecognized OOV words. Additionally, the testing dataset is noisy and relatively small, which may limit the generalizability of the results.

To address these limitations, we will expand the dataset to include more diverse accents and languages, and to explore the use of other post-processing algorithms and machine learning techniques to improve the accuracy of speech recognition systems.

Point 3:

  1. Perhaps, the authors could discuss more about the comparison with some traditional methods to highlight the advantages and contributions of the proposed method.

Response 3: 

As we were the first ones to try combining distance algorithms with commonly used speech recognition systems such as Google Speech-to-text, Vosk, SpeechBrain, QuartzNet, and Wav2Vec, we were unable to compare our results to other studies as we used our own test set for evaluation. Nevertheless, our findings are consistent with those of other research that have similarities to our approach.

Section: 5. Conclusion (new version)

 In this work, we demonstrate the effectiveness of using a statistical approach to reduce errors in speech recognition systems. The Levenshtein and Damerau algorithms were compared by varying a threshold to minimize the WER on MoroccanFrench test set. The results showed that the Damerau algorithm was more effective than the Levenshtein algorithm in reducing the WER, with a threshold of max=0.8.

Our results are in line with other studies [14], [22] that have reported the effective-ness of post-processing methods in reducing errors for recognizing OOV words in ASR systems.

The post-processing method was used to detect and correct OOV words in the text after the speech recognition task. In terms of time consumption, the post-processing took only 0.029 seconds. In terms of WER reduction, it was between 1% and 4%, which indicates the method succeeded in reducing the error without modifying the ASR system itself. However, there is still room for improvement in this method.

In addition to the points mentioned previously, it is worth noting that the findings of this research have important theoretical and practical implications for the field of speech recognition. Theoretically, this research demonstrates the effectiveness of using statistical approaches to reduce errors in off-the-shelf speech recognition systems, which can inform future research in this area. Practically, the results of this research have potential applications in real-world scenarios such as speech-controlled medical assistant robots that can be implemented in retirement homes.

However, it is important to note that there are limitations to this research. One limitation is that the dataset used for evaluation has a relatively low vocabulary of possible misrecognized OOV words. Additionally, the testing dataset is noisy and relatively small, which may limit the generalizability of the results.

To address these limitations, we will expand the dataset to include more diverse accents and languages, and to explore the use of other post-processing algorithms and machine learning techniques to improve the accuracy of speech recognition systems.

Point 4:

  1. The manuscript should be proof read, otherwise it is difficult to understand and read. Please re-check the grammar and spelling.

Response 4: 

we have attempted to clarify the text in all sections by making corrections.

References

  1. Pollmann, C. Ruff, K. Vetter, and G. Zimmermann, “Robot vs. voice assistant: Is playing with pepper more fun than playing with alexa?,” ACM/IEEE International Conference on Human-Robot Interaction, pp. 395–397, Mar. 2020, doi: 10.1145/3371382.3378251. 
  2. ​“How to Build Domain Specific Automatic Speech Recognition Models on GPUs | NVIDIA Developer Blog.” https://developer.nvidia.com/blog/how-to-build-domain-specific-automatic-speech-recognition-models-on-gpus/ (accessed Nov. 29, 2021). 
  3. ​T. Desot, F. Portet, and M. Vacher, “End-to-End Spoken Language Understanding: Performance analyses of a voice command task in a low resource setting,” Comput Speech Lang, vol. 75, p. 101369, Sep. 2022, doi: 10.1016/J.CSL.2022.101369. 
  4. ​J. Kim and P. Kang, “K-Wav2vec 2.0: Automatic Speech Recognition based on Joint Decoding of Graphemes and Syllables,” arXiv:2110.05172v1, 2021. 
  5. ​A. Laptev, A. Andrusenko, I. Podluzhny, A. Mitrofanov, I. Medennikov, and Y. Matveev, “Dynamic Acoustic Unit Augmentation with BPE-Dropout for Low-Resource End-to-End Speech Recognition,” Sensors 2021, Vol. 21, Page 3063, vol. 21, no. 9, p. 3063, Apr. 2021, doi: 10.3390/S21093063. 
  6. ​A. Y. Andrusenko and A. N. Romanenko, “Improving out of vocabulary words recognition accuracy for an end-to-end Russian speech recognition system,” Scientific and Technical Journal of Information Technologies, Mechanics and Optics, vol. 22, no. 6, pp. 1143–1149, Nov. 2022, doi: 10.17586/2226-1494-2022-22-6-1143-1149. 
  7. ​J. v. Lochter, R. M. Silva, and T. A. Almeida, “Multi-level out-of-vocabulary words handling approach,” Knowl Based Syst, vol. 251, Sep. 2022, doi: 10.1016/J.KNOSYS.2022.108911. 
  8. ​F. Y. Putri, D. Hoesen, and D. P. Lestari, “Rule-Based Pronunciation Models to Handle OOV Words for Indonesian Automatic Speech Recognition System,” Proceeding - 2019 5th International Conference on Science in Information Technology: Embracing Industry 4.0: Towards Innovation in Cyber Physical System, ICSITech 2019, pp. 246–251, Oct. 2019, doi: 10.1109/ICSITECH46713.2019.8987472.
  9. Errattahi, A. El Hannani, and H. Ouahmane, “Automatic speech recognition errors detection and correction: A review,” Procedia Computer Science, vol. 128, pp. 32–37, 2018, doi: 10.1016/J.PROCS.2018.03.005.
  10. ​ Fadel, I. Araf, T. Bouchentouf, P.-A. Buvet, F. Bourzeix, and O. Bourja, “Which French speech recognition system for assistant robots?,” pp. 1–5, Mar. 2022, doi: 10.1109/IRASET52964.2022.9737976. 
  11. Subramanyam, “Improving Speech Recognition Accuracy Using Levenshtein Distance algorithm”.
  12. Salimbajevs, “Using sub-word n-gram models for dealing with OOV in large vocabulary speech recognition for Latvian”.
  13. Y. Putri, D. Hoesen, and D. P. Lestari, “Rule-Based Pronunciation Models to Handle OOV Words for Indonesian Automatic Speech Recognition System,” Proceeding - 2019 5th International Conference on Science in Information Technology: Embracing Industry 4.0: Towards Innovation in Cyber Physical System, ICSITech 2019, pp. 246–251, Oct. 2019, doi: 10.1109/ICSITECH46713.2019.8987472.
  14. Réveil, K. Demuynck, and J. P. Martens, “An improved two-stage mixed language model approach for handling out-of-vocabulary words in large vocabulary continuous speech recognition,” Comput Speech Lang, vol. 28, no. 1, pp. 141–162, Jan. 2014, doi: 10.1016/J.CSL.2013.04.003. 
  15. ​[12] Pellegrini and I. Trancoso, “Error detection in broadcast news ASR using Markov chains,” Lecture Notes in Computer Science (including subseries Lecture Notes in Artificial Intelligence and Lecture Notes in Bioinformatics), vol. 6562 LNAI, pp. 59–69, 2011, doi: 10.1007/978-3-642-20095-3_6/COVER. 
  16. Sarma and D. D. Palmer, “Context-based Speech Recognition Error Detection and Correction”.
  17. Bassil and P. Semaan, “ASR Context-Sensitive Error Correction Based on Microsoft N-Gram Dataset,” Mar. 2012, doi: 10.48550/arxiv.1203.5262.
  18. Yang, Y. Li, J. Wang, and Z. Tang, “Post Text Processing of Chinese Speech Recognition Based on Bidirectional LSTM Networks and CRF,” Electronics 2019, Vol. 8, Page 1248, vol. 8, no. 11, p. 1248, Oct. 2019, doi: 10.3390/ELECTRONICS8111248.
  19. “(2) (PDF) Post-editing and Rescoring of ASR Results with Edit Operations Tagging.” https://www.researchgate.net/publication/351687519_Post-editing_and_Rescoring_of_ASR_Results_with_Edit_Operations_Tagging (accessed Jul. 25, 2022).
  20. Zhang, M. Lei, and Z. Yan, “Automatic Spelling Correction with Transformer for CTC-based End-to-End Speech Recognition,” Mar. 2019, doi: 10.48550/arxiv.1904.10045.
  21. D. Emiru, S. Xiong, Y. Li, A. Fesseha, and M. Diallo, “Improving amharic speech recognition system using connectionist temporal classification with attention model and phoneme-based byte-pair-encodings,” Information (Switzerland), vol. 12, no. 2, pp. 1–22, Feb. 2021, doi: 10.3390/INFO12020062.
  22. Y. Andrusenko and A. N. Romanenko, “Improving out of vocabulary words recognition accuracy for an end-to-end Russian speech recognition system,” Scientific and Technical Journal of Information Technologies, Mechanics and Optics, vol. 22, no. 6, pp. 1143–1149, Nov. 2022, doi: 10.17586/2226-1494-2022-22-6-1143-1149. 
  23. ​R. M. Silva, J. v. Lochter, T. A. Almeida, and A. Yamakami, “FastContext: Handling Out-of-Vocabulary Words Using the Word Structure and Context,” Lecture Notes in Computer Science (including subseries Lecture Notes in Artificial Intelligence and Lecture Notes in Bioinformatics), vol. 13654 LNAI, pp. 539–557, 2022, doi: 10.1007/978-3-031-21689-3_38. 
  24. ​J. Zhao et al., “The THUEE System Description for the IARPA OpenASR21 Challenge,” Proceedings of the Annual Conference of the International Speech Communication Association, INTERSPEECH, vol. 2022-September, pp. 4855–4859, 2022, doi: 10.21437/INTERSPEECH.2022-269. 
  25. L. Aouragh, A. Yousfi, U. Sidi, M. Ben Abdellah, and G. Hicham, “Adapting the Levenshtein Distance to Contextual Spelling Correction,” International Journal of Computer Science and Applications, ÓTechnomathematics Research Foundation, vol. 12, no. 1, pp. 127–133, 2015.

Round 2

Reviewer 2 Report

Still there are language and style inaccuracy, for example in lines 13, 113,154, 383, 384. In lines 166 - 171 repeated text.

Eq. 1 is not completed.

Author Response

Response to Reviewer 2 Comments

Point 1:
Still there are language and style inaccuracy, for example in lines 13, 113,154, 383, 384. In lines 166 - 171 repeated text.

Response 1: 

Thank you for your insightful feedback. We have made efforts to revise and enhance the text, with a particular focus on the areas you highlighted.

Line 13: Current speech recognition systems with fixed vocabulary have difficulties recognizing out-of-vocabulary words (OOVs), such as proper nouns and new words. This leads to misunderstandings or even failure in dialog systems.

Line 113: This article [15] addresses the problem of detecting errors in automatic transcriptions using statistical tools. The Markov chain model's ability to model temporal sequences is tested against a Gaussian Mixture Model and a maximum entropy model. Results show that the Markov chain model outperforms the other two, with a 16.7% (Character Error Rate) CER and 860 errors correctly detected. The article suggests that the choice of using a Markov chain or maximum entropy model depends on the application.

Line 154: This paper [21] presents a subword modeling method for phoneme-level OOV words recognition in Amharic, using grapheme-to-phoneme conversion, syllabification for epenthesis, and subword-based decoding. The end-to-end models were trained and evaluated with a 22 hour speech dataset and a 5k testing dataset. The experiment results showed that phoneme-based BPE system with a syllabification algorithm was effective in achieving minimum WER (18.42%) in the CTC-attention end-to-end method.

Line 383, 384: corrected.

Line 166-171: corrected (repeated text).

Point2: Eq. 1 is not completed.

Response 2:  corrected as illustrated in the figure below.

Are the results clearly presented?

We have also improved the results by comparing them with other studies.

Our results align with other studies [14], [22] that have demonstrated the effectiveness of post-processing methods in reducing errors for recognizing out-of-vocabulary (OOV) words in automatic speech recognition (ASR) systems. However, in contrast, our experiment using Ali et al.'s method [30] with different scoring functions showed poor results. Utilizing a dictionary with only one misspelled word and its correct form, we discovered that the method is unable to handle various types of errors within a single word, such as those caused by different accents. In comparison, our method improves upon this by pairing each correctly spelled OOV word with its potential misspelled versions, allowing for a more accurate matching of misspelled OOV words to their correct form, a task that proves difficult for Ali et al.'s method. Additionally, machine learning techniques generally need a large amount of training data to correct with high accuracy the misspelled OOV word [18, 20, 21] and constructing training data by handwork is costly. As compared with them, our method can be realized with low cost. Besides, our method does not depend on particular speech recognizers [20, 21, 22].

Reviewer 4 Report

The authors corrected the manuscript satisfactorily. I recommend acceptance.

Author Response

Thank you for the effort you took to review our manuscript. Your detailed and insightful comments have greatly helped us in improving the manuscript.Thank you again.

Reviewer 5 Report

some updates were done on the paper.

However, the paper still needs modifications.

The authors are not comparing their work with existing research.

Also, there are many recent papers that can be added to teh literature. The literature needs to be more extensive.

Author Response

Response to Reviewer 5 Comments

Point 1:

some updates were done on the paper.

However, the paper still needs modifications.

The authors are not comparing their work with existing research.

Thank you for your feedback, it helps us a lot to improve our manuscript.

Response 1: 

We have attempted to compare our approach with other studies in the Results and Discussion section

Our results align with other studies [14], [22] that have demonstrated the effectiveness of post-processing methods in reducing errors for recognizing out-of-vocabulary (OOV) words in automatic speech recognition (ASR) systems. However, in contrast, our experiment using Ali et al.'s method [30] with different scoring functions showed poor results. Utilizing a dictionary with only one misspelled word and its correct form, we discovered that the method is unable to handle various types of errors within a single word, such as those caused by different accents. In comparison, our method improves upon this by pairing each correctly spelled OOV word with its potential misspelled versions, allowing for a more accurate matching of misspelled OOV words to their correct form, a task that proves difficult for Ali et al.'s method. Additionally, machine learning techniques generally need a large amount of training data to correct with high accuracy the misspelled OOV word [18, 20, 21] and constructing training data by handwork is costly. As compared with them, our method can be realized with low cost. Besides, our method does not depend on particular speech recognizers [20, 21, 22].

Point 2:

Also, there are many recent papers that can be added to teh literature. The literature needs to be more extensive.

Response 2: 

For the literature review, we have added four more studies for the statistical approach; however, I am having difficulty finding very recent work using statistical approaches for improving OOV in recent off-the-shelf speech recognition systems.

The paper [26] presents a technique to improve ASR accuracy by using phonetic distance and domain knowledge to post-process the results of off-the-shelf speech recognition services. The authors use open-source Sphinx-based language models to decrease the WER of the Google speech recognition system. Results show significant improvement over Google ASR and open-source ASR on various corpora, mainly from human-robot interaction.

The study conducted by Traum et al. [27] utilized a method of combining the re-sults of both Google and Sphinx ASR services in order to achieve both general and do-main-specific outcomes.

The authors [28] propose a language model that is trained using a discriminative method and utilizes dependency parsing, which allows for the utilization of long-distance structural features within sentences. The training process is done on a list of the top-scoring possibilities (n-best lists) using the perceptron algorithm. The model is then evaluated by reordering the top possibilities generated by recognizing speech from the Fisher dataset, which contains informal telephone conversations. The results indicate that this approach of training with syntactic features using perceptron-based methods can lead to a decrease in the WER.

The paper [29] describes a new approach to error correction in automatic speech recognition called SoftCorrect. SoftCorrect uses a soft error detection mechanism to avoid the limitations of both explicit and implicit error detection methods. It uses a probability produced by a dedicatedly designed language model to detect whether a token is correct or not. Experiments show that SoftCorrect outperforms previous works by a large margin, achieving 26.1% and 9.4% CER reduction respectively, while still enjoying fast speed of parallel generation.

References

  1. Twiefel, T. Baumann, S. Heinrich, and S. Wermter, “Improving Domain-independent Cloud-Based Speech Recognition with Domain-Dependent Phonetic Post-Processing,” Proceedings of the AAAI Conference on Artificial Intelligence, vol. 28, no. 1, pp. 1529–1535, Jun. 2014, doi: 10.1609/AAAI.V28I1.8929.
  2. Traum, K. Georgila, R. Artstein, and A. Leuski, “Evaluating Spoken Dialogue Processing for Time-Offset Interaction,” SIGDIAL 2015 - 16th Annual Meeting of the Special Interest Group on Discourse and Dialogue, Proceedings of the Conference, pp. 199–208, 2015, doi: 10.18653/V1/W15-4629.
  3. Byambakhishig, E., Tanaka, K., Aihara, R., Nakashika, T., Takiguchi, T., Ariki, Y.: Error correction of automatic speech recognition based on normalized web distance. In: Proceedings of the INTERSPEECH 2014, pp. 2852–2856 (2014)
  4. Leng et al., “SoftCorrect: Error Correction with Soft Detection for Automatic Speech Recognition,” Dec. 2022, doi: 10.48550/arxiv.2212.01039.

Point 3:

Are the methods adequately described?

Response 3:

For the methodology section, we have added more detail to the description of our method.

After speech recognition, we tokenized the input text into individual words (tokens). We then compared each word to the out-of-vocabulary (OOV) words that were misrecognized. To determine how similar the words were, we used Levenshtein dis-tance. If the similarity score between two words was low, it means that the word in the text was detected as an OOV and we would then replace it with the correct OOV word.

The proposed method is a post-correction technique for the outputs of speech recognition systems. It utilizes a combination of dictionary lookup and the Levenshtein distance measure to correct any potential misspellings. The method consists of three steps:

  • Dictionary Creation: The first step is to create a dictionary that includes both the mispelled words and their corresponding correct word, with the correct word being stored as keyword in the dictionary.
  • Searching for a Match: In this step, the word of interest (the output from the ASR) is searched within the dictionary values. If the word is found, it is considered a misspelling. If not, it is considered to be spelled correctly.
  • Calculation of Levenshtein Distance: If a misspelling is identified in the previous step, the word of interest is then subjected to the Levenshtein distance measure. This measure calculates the distance between the misspelled word and all words in the dictionary. The correct word is chosen as the keyword with the smallest Le-venshtein distance, as this indicates the highest similarity between the two words.

The goal of correction is to replace any inaccurate word with its closest match from the dictionary. Our method utilizes traditional techniques for determining the similarity between strings of text, such as Levenshtein distance. To evaluate the performance of this approach, we recalculate the Word Error Rate (WER) of ASR systems using this method, as illustrated in Figure 1.
